# Accelerating Nonconvex Learning via Replica Exchange Langevin Diffusion

**Yi Chen**
Department of Industrial Engineering & Management Science
Northwestern University
Evanston, IL 60201, USA
yichen2016@u.northwestern.edu

**Jinglin Chen**
Department of Computer Science
University of Illinois at Urbana-Champaign
Urbana, IL 61801, USA
jinglinc@illinois.edu

**Jing Dong**
Columbia Business School School
Columbia University
New York City, NY 10027, USA
jing.dong@gsb.columbia.edu

**Jian Peng**
Department of Computer Science
University of Illinois at Urbana-Champaign
Urbana, IL 61801, USA
jianpeng@illinois.edu

**Zhaoran Wang**
Department of Industrial Engineering & Management Science
Northwestern University
Evanston, IL 60201, USA
zhaoran.wang@northwestern.edu

## Abstract

Langevin diffusion is a powerful method for nonconvex optimization, which enables the escape from local minima by injecting noise into the gradient. In particular, the temperature parameter controlling the noise level gives rise to a tradeoff between "global exploration" and "local exploitation", which correspond to high and low temperatures. To attain the advantages of both regimes, we propose to use replica exchange, which swaps between two Langevin diffusions with different temperatures. We theoretically analyze the acceleration effect of replica exchange from two perspectives: (i) the convergence in $\chi^2$-divergence, and (ii) the large deviation principle. Such an acceleration effect allows us to faster approach the global minima. Furthermore, by discretizing the replica exchange Langevin diffusion, we obtain a discrete-time algorithm. For such an algorithm, we quantify its discretization error in theory and demonstrate its acceleration effect in practice.

## 1 Introduction

We consider the problem of minimizing a nonconvex objective function, which arises from numerous machine learning problems such as training neural networks. However, due to the existence of spurious local minima, nonconvex optimization remains challenging in both theory and practice. To overcome this difficulty, one idea is to construct a diffusion process whose invariant distribution concentrates around the global minima (Chiang et al., 1987). If we run such a diffusion process for a sufficiently long time, we expect to draw from its invariant distribution a point that is close to a global minimum with high probability.

One commonly used diffusion process is the Langevin diffusion, which is closely related to first-order optimization (Gidas, 1985). In specific, the Langevin diffusion can be viewed as gradient flow with injected noise, where the noise is scaled by a temperature parameter. Such a temperature parameter gives rise to a tradeoff between two opposite effects, namely "global exploration" and "local

exploitation". More specifically, a higher temperature results in a larger injected noise, which increases the probability of escaping from spurious local minima and facilitates the global exploration of the whole domain. On the other hand, with a lower temperature, the Langevin diffusion behaves more like the gradient flow, which focuses more on exploiting the local geometry to decrease the objective value, yielding a more concentrated invariant distribution.

In this paper, we aim to bridge the gap between "global exploration" and "local exploitation" in nonconvex optimization with replica exchange, which originates from parallel tempering Markov chain Monte Carlo methods for sampling from a multimodal distribution (Earl & Deem, 2005). In contrast to the standard Langevin diffusion driven by a single temperature, the replica exchange Langevin diffusion consists of two Langevin diffusions with low and high temperatures, which locally exploits the geometry and globally explores the domain, respectively. In particular, the two Langevin diffusions exchange their positions following a specific swapping scheme, which ensures that the invariant distribution concentrates around the global minima. See Figures 1.(a)-(b) for an illustration of swapping. Compared with the standard Langevin diffusion, replica exchange accelerates the rate of convergence to the invariant distribution. As a result, with replica exchange we obtain a more accurate solution to the nonconvex optimization problem within a shorter time. It is worth mentioning that, although in this paper we focuses on the case of swapping two Langevin diffusions, our theory and method naturally extend to multiple Langevin diffusions.

Our theory quantifies the acceleration effect of replica exchange from two perspectives. First, we use the theory of Markov semigroup, particularly the Dirichlet form, to characterize the evolution of the $\chi^2$-divergence between the distribution at time $t$ and the invariant distribution. In specific, we show that, compared with the standard Langevin diffusion, replica exchange boosts the Dirichlet form with a nonnegative term, which results in a faster decay of the $\chi^2$-divergence. Further combined with the Poincaré inequality, we establish the exponential rate of convergence of the $\chi^2$-divergence to zero, where the acceleration effect of replica exchange is characterized by the Poincaré constant. We illustrate such an acceleration effect in Figure 1.(c). Second, we use the large deviation principle (LDP) to characterize the exponential decay of the probability that the empirical measure of the sample path deviates from the invariant distribution. In particular, we show that replica exchange boosts the LDP rate function, which implies a faster exponential decay rate. See Figure 1.(d) for an illustration. Both perspectives are built on the infinitesimal generator of the replica exchange Langevin diffusion. We show that essentially the acceleration effects are the consequences of an additional term in the infinitesimal generator, which is introduced by replica exchange. In addition to quantifying the acceleration effect of replica exchange, we use the Euler scheme to discretize the replica exchange Langevin diffusion and propose a discrete-time algorithm. We establish an upper bound of the discretization error via Grönwall's inequality. We also discuss several issues related to the parameters of the replica exchange Langevin diffusion, especially the swapping intensity and temperatures. We defer these discussions to §A of the appendix.

**Related Work:** The idea of nonconvex optimization via diffusion process dates back to simulated annealing (Černỳ, 1985) and is systematically studied in Chiang et al. (1987). For the application of the Langevin diffusion in nonconvex optimization, see, for example, Gidas (1985) and Geman & Hwang (1986) and the followup works. More recently, Dalalyan & Tsybakov (2009); Bubeck et al. (2015); Dalalyan (2017) analyze the nonasymptotic rate of convergence of the Langevin diffusion for strongly convex objective functions. Moreover, Durmus et al. (2017); Zhang et al. (2017); Raginsky et al. (2017); Xu et al. (2018) further extend the analysis to handle nonconvexity and stochastic gradient. See also, for example, Belloni et al. (2015); Ge et al. (2015); Hazan et al. (2016); Cheng et al. (2017); Zou et al. (2018b;a; 2019) for related works on the extensions of the Langevin diffusion and other diffusions. However, all these works focus on the setting with a single temperature. In comparison, we study the acceleration effect of replica exchange, which utilizes more than one temperature. Hence, our work is orthogonal to these existing works and can potentially be applied to accelerate other diffusion processes.

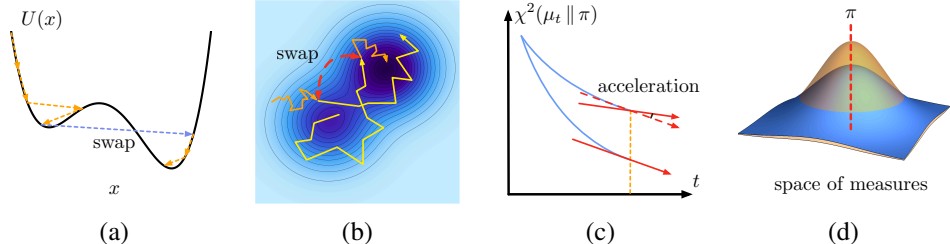

Figure 1: An illustration of the replica exchange Langevin diffusion. In (a) we illustrate the diffusion process driven by the low temperature, which locally exploits the geometry. The orange dashed lines characterize the evolution of the standard Langevin diffusion, while the blue dashed line denotes the swap, which drives the diffusion process into the neighborhood of another local minimum. In (b) we illustrate the trajectories of a pair of diffusion processes driven by two temperatures. The orange and yellow lines correspond to the low and high temperatures, respectively. The black lines are the contours of the objective function. Note that the lines of the same colors are disjoint due to the swap, which is denoted by the red dashed line. In (c) we illustrate the evolution of the $\chi^2$-divergence. The upper and lower blue curves correspond to diffusion processes with zero and positive swapping intensities, respectively. The two solid red arrows denote the derivatives of the $\chi^2$-divergence, and the angle between them characterizes the acceleration effect. In (d) we illustrate the concentration of the empirical measures. The horizontal plane denotes the space of measures, whose center is $\pi$, that is, the stationary distribution of the replica exchange Langevin diffusion. The yellow and blue surfaces that center at $\pi$ characterize the probability densities of the empirical measures, corresponding to zero and positive swapping intensities, respectively. Compared with the blue surface, the yellow one is more concentrated around $\pi$, which also characterizes the acceleration effect of swapping.

In a parallel line of works, the idea of replica exchange is studied in the context of parallel tempering Markov chain Monte Carlo for sampling from a multimodal distribution. See, for example, Earl & Deem (2005); Sindhikara et al. (2008) and the references therein. More recently, Woodard et al. (2009) study the mixing time of parallel tempering, while Dupuis et al. (2012) extend the swapping rate of replica exchange to infinity and establish the corresponding LDP. In addition, another well-studied method is simulated tempering, a technique similar to replica exchange, which also aims at accelerating the convergence of diffusion processes (Marinari & Parisi, 1992; Geyer & Thompson, 1995). The main difference between replica exchange and simulated tempering is that the former tracks multiple diffusion processes driven by various temperatures and accelerates the convergence by swapping, while the latter tracks only one diffusion process but treats the temperature as a stochastic process. See, for example, Zheng (2003) and the references therein for a detailed discussion on the difference. More recently, Ge et al. (2017) study the convergence of simulating tempering for nonconvex objective functions from the perspective of mixing time, which is also orthogonal to our work. The LDP techniques in our work may potentially be applied to the analysis of simulated tempering as well.

**Main Contribution:** Despite the broad application in sampling, to the best of our knowledge, this paper is the first attempt to apply the replica exchange Langevin diffusion in the context of nonconvex optimization. In summary, our contribution is twofold: (i) We quantify the acceleration effect of the replica exchange Langevin diffusion from two perspectives, that is, the convergence of the $\chi^2$-divergence and the LDP for empirical measures. (ii) We propose a nonconvex optimization algorithm based on the discretization of the replica exchange Langevin diffusion and establish an upper bound for the discretization error.

## 2 BASIC IDEA

We consider the following unconstrained optimization problem,

$$\underset{x \in \mathbb{R}^d}{\text{minimize}} \, U(x). \tag{2.1}$$

When $U(x)$ is nonconvex, it is difficult to obtain its global minima. A commonly used algorithm is the Langevin diffusion, which is defined by the stochastic differential equation

$$\mathrm{d}X_t = -\nabla U(X_t)\mathrm{d}t + \sqrt{2\tau}\mathrm{d}W_t. \tag{2.2}$$

Here $\{W_t\}_{t\geq 0}$ is a standard $d$-dimensional Brownian motion and $\tau > 0$ is the temperature parameter. Under mild regularity conditions, the Langevin diffusion $\{X_t\}_{t\geq 0}$ has a unique invariant distribution that is absolutely continuous with respect to the Lebesgue measure with density

$$\pi_\tau(x) = \frac{e^{-U(x)/\tau}}{\int_{\mathbb{R}^d} e^{-U(x)/\tau}\mathrm{d}x}. \tag{2.3}$$

Here $\int_{\mathbb{R}^d} e^{-U(x)/\tau}\mathrm{d}x$ is the normalization constant. In particular, $\pi_\tau$ is the limiting distribution of $\{X_t\}_{t\geq 0}$, which means that with any initialization $X_0$, the distribution of $X_t$ converges to $\pi_\tau$ (Bakry et al., 2013). We observe from (2.3) that with $\tau \to 0$ the probability measure $\pi_\tau$ concentrates around the global minima of $U(x)$ (Hwang, 1980). In other words, if we choose a sufficiently small temperature parameter $\tau$ and run the Langevin diffusion in (2.2) for a sufficiently long time, we expect to obtain a solution that falls into the neighborhood of a global minimum of $U(x)$ with high probability.

Here the temperature parameter $\tau$ plays a crucial role. In practice, we can only run the Langevin diffusion for a finite time, giving rise to a tradeoff between "global exploration" and "local exploitation", which correspond to high and low temperatures. In specific, when the temperature parameter $\tau$ is small, the convergence of $X_t$ is slow and the particle can be trapped by a local minimum for a long time without globally exploring the whole domain. Consequently, within a finite time, we can only obtain a sample from a distribution that is far away from the invariant distribution $\pi_\tau$. In contrast, when the temperature parameter $\tau$ is large, the convergence is accelerated due to better global exploration. The distribution of $X_t$ converges faster to the invariant distribution $\pi_\tau$ globally, but locally $\pi_\tau$ is less concentrated around the global minima of $U(x)$.

To bridge the gap between high and low temperatures, in this paper we study an adaptive algorithm called replica exchange Langevin diffusion. In detail, we consider a pair of particles driven by two Langevin diffusions as defined in (2.2) with temperatures $\tau_1 < \tau_2$, respectively. We use $Z_t = (Z_t^{(1)}, Z_t^{(2)})$ to denote the positions of the two particles at time $t$. In other words, we have

$$\mathrm{d}Z_t^{(1)} = -\nabla U\big(Z_t^{(1)}\big)\mathrm{d}t + \sqrt{2\tau_1}\mathrm{d}W_t^{(1)}, \quad \mathrm{d}Z_t^{(2)} = -\nabla U\big(Z_t^{(2)}\big)\mathrm{d}t + \sqrt{2\tau_2}\mathrm{d}W_t^{(2)}. \tag{2.4}$$

According to (2.3), the invariant distribution of $\{Z_t\}_{t\geq 0}$ is absolutely continuous with respect to the Lebesgue measure on $\mathbb{R}^{2d}$ and its density is proportional to

$$\mu(x_1, x_2) = \exp\big(-U(x_1)/\tau_1 - U(x_2)/\tau_2\big). \tag{2.5}$$

The marginal invariant distribution of the low-temperature particle concentrates around the global minima of $U(x)$. Hence, the low-temperature particle is of particular interest for the purpose of nonconvex optimization. The key idea of replica exchange is to enable the low-temperature particle to achieve better global exploration by swapping its position with the high-temperature particle. In specific, at time $t$ and positions $(x_1, x_2)$, the two particles swap with rate

$$a \cdot s(x_1, x_2) = a \cdot \left(1 \wedge \frac{\mu(x_2, x_1)}{\mu(x_1, x_2)}\right), \tag{2.6}$$

which means

$$\mathbb{P}\big(Z_{t+\mathrm{d}t} = (x_2, x_1) \,|\, Z_t = (x_1, x_2)\big) = a \cdot s(x_1, x_2)\mathrm{d}t,$$
$$\mathbb{P}\big(Z_{t+\mathrm{d}t} = (x_1, x_2) \,|\, Z_t = (x_1, x_2)\big) = 1 - a \cdot s(x_1, x_2)\mathrm{d}t. \tag{2.7}$$

Here $a \geq 0$ is a constant called swapping intensity. As is shown in Lemma 3.2, the specific form of $s(x_1, x_2)$ in (2.6) ensures that for any $a$, the invariant distribution of the replica exchange Langevin diffusion $\{Z_t\}_{t\geq 0}$ is the same as (2.5), that is, as if the two particles are independent. Corresponding to the continuous-time process in (2.4)-(2.6), in §3.4 of the appendix we consider the replica

exchange stochastic gradient descent algorithm, which corresponds to the discretization of $\{Z_t\}_{t \geq 0}$ using Euler scheme.

To better understand the intuition behind replica exchange, note that we use two particles driven by low and high temperatures to achieve "local exploitation" and "global exploration", respectively. For the purpose of optimization, we only need to track the trajectory of the first particle. By plugging (2.5) in (2.6), we obtain

$$a \cdot s(x_1, x_2) = a \cdot \exp\Big( 0 \wedge (1/\tau_1 - 1/\tau_2) \cdot \big( U(x_1) - U(x_2) \big) \Big). \tag{2.8}$$

Since $\tau_1 < \tau_2$, the rate $a \cdot s(x_1, x_2)$ is monotone increasing with respect to the difference between the objective values at $x_1$ and $x_2$, which denote the positions of the two particles. Hence, the two particles are more likely to swap when the first one has a larger objective value. In other words, swapping tends to move the first particle, which we are interested in, to a position corresponding to a lower objective value. An extreme case of the replica exchange Langevin diffusion is $\tau_1 = 0$ and $\tau_2 = \infty$. In this case, the first equation in (2.4) reduces to an ordinary differential equation characterizing the deterministic gradient descent, while the second one in (2.4) corresponds to an approximated uniform exploration of the whole domain $\mathbb{R}^d$. According to (2.8), the particles never swap if the objective value of the first particle is smaller than the second. Hence, roughly speaking, the replica exchange Langevin diffusion reduces to the deterministic gradient descent with uniformly randomized restarts. In contrast to this extreme case, for $0 < \tau_1 < \tau_2 < \infty$, the second particle globally explores the whole domain more adaptively with not only noise but also gradient information. In particular, its stationary distribution has a larger density around the local minima, which leads to better restarts that adapt to the global geometry for the first particle.

## 3 THEORETICAL ANALYSIS

In this section, we lay out the theoretical analysis of replica exchange Langevin diffusion introduced in §2, which demonstrates the acceleration effect of swapping. Due to space constraint, we defer the necessary background and detailed proofs to §B and §C of the appendix, respectively. Throughout the following analysis, we assume that the objective function $U(\cdot)$ satisfies the following assumption.

**Assumption 3.1** (Smoothness and Dissipativity). The function $U(\cdot)$ is $L$-smooth, that is, there exists a positive constant $L$ such that for all $x, y \in \mathbb{R}^d$

$$\big\| \nabla U(x) - \nabla U(y) \big\| \leq L \|x - y\|. \tag{3.1}$$

The function $U(\cdot)$ is also $(\alpha, \beta)$-dissipative, that is, there exist positive constants $\alpha$ and $\beta$ such that for all $x \in \mathbb{R}^d$

$$\big\langle x, \nabla U(x) \big\rangle \geq \alpha \|x\|^2 - \beta. \tag{3.2}$$

The assumption on smoothness in (3.1) characterizes the Lipschitz continuity of the gradient of objective function, which is commonly used in the optimization literature (Nesterov, 2013). The assumption on dissipativity in (3.2), roughly speaking, characterizes the approximate quadratic growth of the objective function at infinity, which is commonly used in the control and dynamic system literature (Hale, 2010). This condition is also used in Raginsky et al. (2017). It is worth noting that our theory does not require the convexity assumption.

### 3.1 INVARIANCE AND REVERSIBILITY

In this section, we show that the invariant distribution of the replica exchange Langevin diffusion $\{Z_t\}_{t \geq 0}$ has the form in (2.5) rigorously, whose first component preserves the concentration. We also show the reversibility of $\{Z_t\}_{t \geq 0}$, which is a nice property necessary for the subsequent convergence analysis. We consider the replica exchange Langevin diffusion $\{Z_t\}_{t \geq 0}$ defined by (2.4)-(2.6), where $Z_t = (Z_t^{(1)}, Z_t^{(2)})$ denotes the positions of the two particles at time $t$, and $a \geq 0$ is

the swapping intensity that controls the frequency of swapping. The following auxiliary lemma characterizes the invariant distribution of $\{Z_t\}_{t\geq 0}$ and its reversibility. Restricted to the space, we provide a detailed proof in §C.1 of the appendix.

**Lemma 3.2.** The infinitesimal generator of $\{Z_t\}_{t\geq 0}$ with swapping intensity $a$, which is defined in (2.4)-(2.6), takes the form

$$\mathscr{L}^a\big(f(x_1, x_2)\big) = \underbrace{-\langle \nabla_{x_1} f(x_1, x_2), \nabla_{x_1} U(x_1)\rangle + \tau_1 \Delta_{x_1} f(x_1, x_2)}_{\mathscr{L}_1^a(f(x_1, x_2))} \tag{3.3}$$

$$\underbrace{-\langle \nabla_{x_2} f(x_1, x_2), \nabla_{x_2} U(x_2)\rangle + \tau_2 \Delta_{x_2} f(x_1, x_2)}_{\mathscr{L}_2^a(f(x_1, x_2))} + \underbrace{a \cdot s(x_1, x_2) \cdot \big(f(x_2, x_1) - f(x_1, x_2)\big)}_{\mathscr{L}_s^a(f(x_1, x_2))}.$$

Moreover, $\{Z_t\}_{t\geq 0}$ is reversible and its invariant distribution $\pi$ has density

$$\mathrm{d}\pi(x_1, x_2) \propto \mu(x_1, x_2)\mathrm{d}x_1\mathrm{d}x_2 \tag{3.4}$$

with respect to the Lebesgue measure on $\mathbb{R}^{2d}$, where $\mu(x_1, x_2)$ is defined in (2.5).

In Lemma 3.2, the first two terms $\mathscr{L}_1^a$ and $\mathscr{L}_2^a$ on the right-hand side of (3.3) correspond to the standard Langevin diffusion, while the last term $\mathscr{L}_s^a$ arises from swapping. The replica exchange Langevin diffusion $\{Z_t\}_{t\geq 0}$ defined in (2.4)-(2.6) is ergodic, which means that, with any initialization, the distribution of $Z_t$ converges to the invariant distribution $\pi$. There are two perspectives to characterize the convergence of the Markov process $\{Z_t\}_{t\geq 0}$: (i) the $\chi^2$-divergence between the distribution of $Z_t$ and the invariant distribution $\pi$, and (ii) the convergence of the empirical measure of $\{Z_t\}_{t\geq 0}$, which is viewed as a random element in the space of measures. In the following, we quantify the convergence of $\{Z_t\}_{t\geq 0}$ from both perspectives. In §3.2, we use Poincaré inequality to quantify (i), while in §3.3, we apply the large deviation principle (LDP) to characterize (ii). In particular, we show that the term $\mathscr{L}_s^a$ in (3.3) plays a crucial role in accelerating the convergence.

## 3.2 CONVERGENCE IN $\chi^2$-DIVERGENCE

Let $\mu_t$ be the distribution of the replica exchange Langevin diffusion $\{Z_t\}_{t\geq 0}$ at time $t$, and $\pi$ be its invariant distribution. In the following, we quantify the discrepancy between $\mu_t$ and $\pi$ using the $\chi^2$-divergence, which is defined as

$$\chi^2(\mu_t \,\|\, \pi) = \int \left(\frac{\mathrm{d}\mu_t}{\mathrm{d}\pi} - 1\right)^2 \mathrm{d}\pi,$$

where $\mathrm{d}\mu_t/\mathrm{d}\pi$ is the Radon-Nikodym derivative between $\mu_t$ and $\pi$. In the following, we characterize the evolution of $\chi^2(\mu_t \,\|\, \pi)$ along time $t$ using the Dirichlet form, which is defined as

$$\mathscr{E}^a(f) = \int \Gamma^a(f)\mathrm{d}\pi, \quad \text{where } \Gamma^a(f) = 1/2 \cdot \big(\mathscr{L}^a(f^2) - 2f\mathscr{L}^a(f)\big). \tag{3.5}$$

Here $\Gamma^a$ is called Carré du Champ operator and recall that we use $\pi$ and $\mathscr{L}^a$ to denote the invariant distribution and infinitesimal generator of $\{Z_t\}_{t\geq 0}$, respectively. In other words, the Dirichlet form is defined as the integration of the Carré du Champ operator under the invariant distribution $\pi$.

To characterize the evolution of the $\chi^2$-divergence, we take the derivative of $\chi^2(\mu_t \,\|\, \pi)$ with respect to time $t$. Recall that $\mu_t$ is the distribution of $Z_t$, and by (B.2) in the appendix the corresponding semigroup $\{P_t\}_{t\geq 0}$ is defined as $P_t(f(x)) = \mathbb{E}[f(Z_t) \,|\, Z_0 = x]$. By setting $f$ as $\mathrm{d}\mu_0/\mathrm{d}\pi$ and the definition of conditional expectation, we have $\mathrm{d}\mu_t/\mathrm{d}\pi = P_t(\mathrm{d}\mu_0/\mathrm{d}\pi)$, which implies

$$\frac{\mathrm{d}}{\mathrm{d}t}\chi^2(\mu_t \,\|\, \pi) = \frac{\mathrm{d}}{\mathrm{d}t}\int \left[P_t\left(\frac{\mathrm{d}\mu_0}{\mathrm{d}\pi}\right)\right]^2 \mathrm{d}\pi = 2\int P_t\left(\frac{\mathrm{d}\mu_0}{\mathrm{d}\pi}\right) \cdot \frac{\mathrm{d}}{\mathrm{d}t}\left[P_t\left(\frac{\mathrm{d}\mu_0}{\mathrm{d}\pi}\right)\right]\mathrm{d}\pi$$

$$= 2\int P_t\left(\frac{\mathrm{d}\mu_0}{\mathrm{d}\pi}\right) \cdot \mathscr{L}^a\left(P_t\left(\frac{\mathrm{d}\mu_0}{\mathrm{d}\pi}\right)\right)\mathrm{d}\pi, \tag{3.6}$$

where the last equation follows from the definition of the infinitesimal generator in Definition B.2 in the appendix. Meanwhile, by Definition B.3 in the appendix and the invariance of $\pi$, we have $\int \mathscr{L}^a(f^2)\mathrm{d}\pi = 0$, which together with (3.5) implies

$$\mathscr{E}^a(f) = -\int f\mathscr{L}^a(f)\mathrm{d}\pi. \tag{3.7}$$

Combining (3.6) and (3.7) we obtain

$$\frac{\mathrm{d}}{\mathrm{d}t}\chi^2(\mu_t \,\|\, \pi) = -2\mathscr{E}^a\left(P_t\left(\frac{\mathrm{d}\mu_0}{\mathrm{d}\pi}\right)\right) = -2\mathscr{E}^a\left(\frac{\mathrm{d}\mu_t}{\mathrm{d}\pi}\right), \tag{3.8}$$

which shows that the derivative of the $\chi^2$-divergence between $\mu_t$ and the invariant distribution $\pi$ is exactly negative twice the Dirichlet form of the Radon-Nykodim derivative $\mathrm{d}\mu_t/\mathrm{d}\pi$. In the following theorem, we show that a larger swapping intensity $a$ boosts the Dirichlet form and thus accelerates the convergence. Recall that, as defined in (3.5), $\mathscr{E}^a$ is the Dirichlet form of the replica exchange Langevin diffusion $\{Z_t\}_{t\geq 0}$ with swapping intensity $a$.

**Theorem 3.3.** For any fixed function $f$, $\mathscr{E}^a(f)$ is an increasing nonnegative function with respect to $a$. In particular, we have

$$\mathscr{E}^a(f) = \underbrace{\int \left(\tau_1 \cdot \left\|\nabla_{x_1} f(x_1, x_2)\right\|^2 + \tau_2 \cdot \left\|\nabla_{x_2} f(x_1, x_2)\right\|^2\right)\mathrm{d}\pi(x_1, x_2)}_{\mathscr{E}^0(f)}$$

$$+ \underbrace{\int a/2 \cdot s(x_1, x_2) \cdot \left(f(x_2, x_1) - f(x_1, x_2)\right)^2 \mathrm{d}\pi(x_1, x_2)}_{\text{Acceleration Effect}}, \tag{3.9}$$

where the both terms on the right-hand side are nonnegative.

See §C.2 of the appendix for a detailed proof of Theorem 3.3. Combined with (3.8), Theorem 3.3 shows that swapping accelerates the evolution of $\chi^2(\mu_t \,\|\, \pi)$. In particular, the Dirichlet form $\mathscr{E}^a$ on the right-hand side of (3.8) decomposes into two terms. The first term $\mathscr{E}^0$ in (3.9) corresponds to the replica exchange Langevin diffusion without swapping, that is, $a = 0$. More specifically, the two nonnegative terms in $\mathscr{E}^0$ characterize the individual dynamics of the two particles, respectively. In particular, larger temperatures $\tau_1$ and $\tau_2$ yield a larger $\mathscr{E}^0$, which implies a faster rate of convergence. Meanwhile, as shown in the proof of Theorem 3.3, the specific form of swapping in (2.7) ensures the nonnegativity of the second term in (3.9), which further boosts the Dirichlet form and leads to faster evolution of $\chi^2(\mu_t \,\|\, \pi)$ in (3.8). It is worth mentioning that Theorem 3.3 does not rely on Assumption 3.3 and holds for all objective functions $U(\cdot)$.

To better understand Theorem 3.3, note that the symmetry of $f$, which corresponds to $\mathrm{d}\mu_t/\mathrm{d}\pi(x_1, x_2)$ in (3.8), plays a crucial role in the acceleration effect. In specific, when $\mathrm{d}\mu_t/\mathrm{d}\pi(x_1, x_2)$ is asymmetric, the second term of the Dirichlet form in (3.9) is positive. Otherwise when $\mathrm{d}\mu_t/\mathrm{d}\pi(x_1, x_2)$ is symmetric, the second term in (3.9) vanishes, since $\mathrm{d}\mu_t/\mathrm{d}\pi(x_1, x_2) - \mathrm{d}\mu_t/\mathrm{d}\pi(x_2, x_1) = 0$ for all $x_1$ and $x_2$. In other words, the acceleration effect degenerates due to symmetry. Intuitively, at time $t$ the two particles are equivalent and swapping does not change their joint distribution, and hence does not affect the convergence of $\chi^2(\mu_t \,\|\, \pi)$.

Based on (3.8), we further establish the exponential convergence of $\chi^2(\mu_t \,\|\, \pi)$. The key ingredient is the Poincaré inequality (Bakry et al., 2013). Furthermore, we show that such an exponential rate of convergence is dictated by the constant in the Poincaré inequality, which is called the Poincaré constant. In particular, we show how the swapping intensity $a$ affects the Poincaré constant and accelerates the exponential convergence.

For a Markov process, the Poincaré inequality states that the $\chi^2$-divergence between any probability measure and its invariant distribution is uniformly upper bounded by the Dirichlet form of the Radon-Nykodim derivative, provided that the probability measure of interest is absolutely continuous with respect to the invariant distribution. In specific, we say that $\{Z_t\}_{t\geq 0}$ satisfies the Poincaré

inequality with Poincaré constant $\kappa$, if for all probability measures $\nu \ll \pi$, the following inequality holds,

$$\chi^2(\nu \,\|\, \pi) \leq \kappa \cdot \mathscr{E}^a\left(\frac{\mathrm{d}\nu}{\mathrm{d}\pi}\right). \tag{3.10}$$

Note that the Poincaré inequality is specified by the invariant distribution $\pi$ and the Dirichlet form $\mathscr{E}^a$.

When the Poincaré inequality in (3.10) holds, the derivative of the $\chi^2$-divergence in (3.8) is upper bounded by itself, which implies the exponential rate of convergence. In particular, we have

$$\frac{\mathrm{d}}{\mathrm{d}t}\chi^2(\mu_t \,\|\, \pi) \leq -2\kappa^{-1} \cdot \chi^2(\mu_t \,\|\, \pi), \ \text{ which yields } \ \chi^2(\mu_t \,\|\, \pi) \leq \chi^2(\mu_0 \,\|\, \pi) \cdot e^{-2t/\kappa}. \tag{3.11}$$

Establishing the Poincaré inequality is highly nontrivial. However, for the replica exchange Langevin diffusion, the following theorem shows that Poincaré inequality holds under the smoothness and dissipativity conditions in Assumption 3.1, which implies (3.11).

**Theorem 3.4.** Under Assumption 3.1, the replica exchange Langevin diffusion specified in (2.4)-(2.6) satisfies the Poincaré inequality in (3.10). Hence, the second inequality in (3.11) holds, which implies the exponential decay of $\chi^2$-divergence.

The proof of Theorem 3.4 adapts from the proof in Bakry et al. (2008), where a standard Langevin diffusion with unique driving temperature is considered. See §C.3 of the appendix for a detailed proof.

### 3.3 LARGE DEVIATION PRINCIPLE (LDP) ANALYSIS

We focus on the empirical measure of the replica exchange Langevin diffusion $\{Z_t\}_{t\geq 0}$ defined in (2.4)-(2.6), which is in the same spirit of Polyak averaging (Polyak & Juditsky, 1992) but replaces the iterates with Dirac measures. In specific, for any fixed time $T > 0$, the empirical measure of $\{Z_t\}_{t\geq 0}$ is defined as

$$\nu_T = \frac{1}{T}\int_0^T \delta_{Z_t}\mathrm{d}t, \tag{3.12}$$

where $\delta_{Z_t}$ denotes the Dirac measure at $Z_t$. Note that $\{\nu_T\}_{T\geq 0}$ is a sequence of random measures, which are random elements of $\mathcal{P}(\mathbb{R}^{2d})$, that is, the space of probability measures on $\mathbb{R}^{2d}$. We equip $\mathcal{P}(\mathbb{R}^{2d})$ with the topology of weak convergence, which enables us to define open and closed sets of $\mathcal{P}(\mathbb{R}^{2d})$.

Recall that Lemma 3.2 shows that the replica exchange Langevin diffusion $\{Z_t\}_{t\geq 0}$ is reversible. This fact enables us to apply the Donsker-Varadhan theory (Donsker & Varadhan, 1975), which implies that $\{Z_t\}_{t\geq 0}$ obeys LDP. Formally, we have the following theorem.

**Theorem 3.5.** The sequence of empirical measures $\{\nu_T\}_{T\geq 0}$ of the replica exchange Langevin diffusion $\{Z_t\}_{t\geq 0}$ defined in (3.12) obeys LDP. That is to say, for all open sets $\mathcal{O}$ and closed sets $\mathcal{F}$ in $\mathcal{P}(\mathbb{R}^{2d})$, the following inequalities hold,

$$\liminf_{T\to\infty}\frac{1}{T}\log\mathbb{P}(\nu_T \in \mathcal{O}) \geq -\inf_{\nu\in\mathcal{O}}I^a(\nu), \quad \limsup_{T\to\infty}\frac{1}{T}\log\mathbb{P}(\nu_T \in \mathcal{F}) \leq -\inf_{\nu\in\mathcal{F}}I^a(\nu).$$

Here $I^a(\cdot) : \mathcal{P}(\mathbb{R}^{2d}) \to [0,\infty]$ is the rate function, which takes the form

$$I^a(\nu) = \underbrace{\int \tau_1 \cdot \left\|\nabla_{x_1}\sqrt{\mathrm{d}\nu/\mathrm{d}\pi(x_1,x_2)}\right\|^2 + \tau_2 \cdot \left\|\nabla_{x_2}\sqrt{\mathrm{d}\nu/\mathrm{d}\pi(x_1,x_2)}\right\|^2 \mathrm{d}\pi(x_1,x_2)}_{I^0(\nu)}$$

$$+ \underbrace{\int a/2 \cdot s(x_1,x_2) \cdot \left(\sqrt{\mathrm{d}\nu/\mathrm{d}\pi(x_2,x_1)} - \sqrt{\mathrm{d}\nu/\mathrm{d}\pi(x_1,x_2)}\right)^2 \mathrm{d}\pi(x_1,x_2)}_{\text{Acceleration Effect}} \tag{3.13}$$

for all $\nu \ll \pi$, and $I^a(\nu) = \infty$ otherwise.

See §C.5 of the appendix for a detailed proof of theorem 3.5. The LDP rate function $I^a(\cdot)$ in Theorem 3.5 characterizes the rate of convergence for the empirical measures $\{\nu_T\}_{T \geq 0}$ towards its population version, which is the invariance distribution of $\{Z_t\}_{t \geq 0}$. More specific, it quantifies the exponential rate of decay of the probability that empirical empirical measures $\{\nu_T\}_{T \geq 0}$ deviate from the invariant distribution $\pi$. Recall that, under weak topology, $\mathcal{P}(\mathbb{R}^{2d})$ is metrizable via the Lévy-Prokhorov metric $d_{\mathrm{LP}}(\cdot, \cdot)$ (Billingsley, 2013). We use $\mathcal{B}_r$ to denote the open ball centered at $\pi$ with radius $r$ in the metric space $(\mathcal{P}(\mathbb{R}^{2d}), d_{\mathrm{LP}})$. Then $\mathcal{B}_r^c$ is a closed set and $\bar{\mathcal{B}}_r^c$ is open. Then according to Theorem 3.5, we have

$$\liminf_{T \to \infty} \frac{1}{T} \log \mathbb{P}\big(d_{\mathrm{LP}}(\nu_T, \mu) > r\big) = \liminf_{T \to \infty} \frac{1}{T} \log \mathbb{P}(\nu_T \in \bar{\mathcal{B}}_r^c) \geq -\inf_{\nu \in \bar{\mathcal{B}}_r^c} I(\nu),$$

$$\limsup_{T \to \infty} \frac{1}{T} \log \mathbb{P}\big(d_{\mathrm{LP}}(\nu_T, \mu) \geq r\big) = \limsup_{T \to \infty} \frac{1}{T} \log \mathbb{P}(\nu_T \in \mathcal{B}_r^c) \leq -\inf_{\nu \in \mathcal{B}_r^c} I(\nu).$$

If we ignore the slight difference between $\mathcal{B}_r^c$ and $\bar{\mathcal{B}}_r^c$, when $T$ is large, the probability $\mathbb{P}(d_{\mathrm{LP}}(\nu_T, \mu) \geq r)$ has approximated scale of $\exp(-T \inf_{\nu \in \mathcal{B}_r^c} I(\nu))$. That is to say, $\inf_{\nu \in \mathcal{B}_r^c} I(\nu)$ characterizes the exponential rate of decay of the probability $\mathbb{P}(d_{\mathrm{LP}}(\nu_T, \mu) \geq r)$ as $T$ goes to infinity. A larger rate function implies a faster exponential rate of decay. Note that LDP includes both upper and lower bounds in Theorem 3.5 and hence, the exponential rate of convergence characterized by LDP is tight.

To see how swapping accelerates the convergence of the empirical measure towards its invariant distribution, note that the LDP rate function in (3.13) decomposes into two terms. The first term corresponds to the LDP rate function of replica exchange Langevin diffusion without swapping, that is, $a = 0$. In specific, it is the sum of two nonnegative terms, which characterize the individual dynamics of the two particles, respectively. Meanwhile, the second term in (3.13) is nonnegative and boosts the LDP rate function, which accelerates the convergence of the empirical measure. Similar to the convergence of the $\chi^2$-divergence in Theorem 3.3, a larger swapping intensity $a$ yields a more significant acceleration effect. The acceleration effect also degenerates when the Radon-Nykodim derivative $\mathrm{d}\nu/\mathrm{d}\pi$ is symmetric. Finally, it is worth noting that, like Theorem 3.3, Theorem 3.5 does not rely on Assumption 3.3 and holds for all objective functions $U(\cdot)$. In summary, Theorems 3.3 and 3.5 together justify the benefit of swapping from two different perspectives.

Note that the LDP rate function given in Theorem 3.5 takes a different form compared with the one obtained in Dupuis et al. (2012). Despite the different the forms, the two results are not contradictory. Our formula is derived from the Donsker-Varadhan theory, which uses the assumption of reversibility, while theirs is obtained from a dual formulation of the LDP rate function. Moreover, one can show that they are equivalent.

## 3.4 DISCRETIZATION ERROR ANALYSIS

In this section, we lay out the discretization error analysis of replica exchange Langevin diffusion. Our previous discussion in §3.2 and §3.3 focuses on the continuous-time process $\{Z_t\}_{t \geq 0}$ defined in (2.4)-(2.6), which needs to be discretized in practice. However, swapping the positions of particles makes it hard to analyze the discretization error, since there is no way to incorporate the swapping of positions into one unified stochastic differential equation. To overcome this difficulty, note that the two particles in the replica exchange Langevin diffusion swap their positions with a specific rate, which is equivalent in distribution to swapping their temperatures with the same rate (Dupuis et al., 2012). We name the later equivalent process as the temperature swapping Langevin diffusion. With a slight abuse of notations, we still use $\{Z_t\}_{t \geq 0}$ to denote the positions of the two particles in the temperature swapping Langevin diffusion. In specific, $\{Z_t\}_{t \geq 0}$ is characterized by the following stochastic differential equation,

$$\mathrm{d}Z_t = -\nabla U(Z_t)\mathrm{d}t + \Sigma_t \mathrm{d}W_t. \tag{3.14}$$

Here $\Sigma_t$ is a random matrix that switches between the diagonal matrices $\mathrm{diag}\{\sqrt{2\tau_1} \cdot I_d, \sqrt{2\tau_2} \cdot I_d\}$ and $\mathrm{diag}\{\sqrt{2\tau_2} \cdot I_d, \sqrt{2\tau_1} \cdot I_d\}$, where $I_d$ denotes the $d$-dimensional identity matrix. The matrix $\Sigma_t$

characterizes the switching of temperatures, and the rate of switching is same as the one specified in (2.6) and (2.7). Note that the temperature swapping Langevin diffusion in (3.14) is equivalent to the replica exchange Langevin diffusion defined in (2.4)-(2.6) in distribution. Furthermore, the stochastic differential equation in (3.14) enables us to define a continuous-time interpolation of the discrete-time sequence, which is useful in the discretization error analysis. Hence, for the purpose of optimization, we only need to discretize and analyze the temperature swapping Langevin diffusion in (3.14).

Note that (3.14) defines a Markov jump diffusion, which can be discretized using the Euler scheme. We denote by $(Z^{(1)}(k), Z^{(2)}(k))$ and $(\tau^{(1)}(k), \tau^{(2)}(k))$ the positions and temperatures of the two particles at discrete time $k$, respectively. Let $(Z^{(1)}(0), Z^{(2)}(0)) = Z_0$ and $(\tau^{(1)}(0), \tau^{(2)}(0)) = (\tau_1, \tau_2)$ be the initialization. For all integers $k \geq 0$, we update the positions of the two particles sequentially as following,

$$Z^{(i)}(k+1) = Z^{(i)}(k) - \eta \cdot \nabla U\big(Z^{(i)}(k)\big) + \sqrt{2\eta \cdot \tau^{(i)}(k)} \cdot \xi^{(i)}(k), i = 1, 2 \qquad (3.15)$$

and swap temperatures according to the following rule,

$$\big(\tau^{(1)}(k+1), \tau^{(2)}(k+1)\big) = \big(\tau^{(2)}(k), \tau^{(1)}(k)\big) \text{ with probability } a \cdot \eta \cdot s\big(Z^{(1)}(k), Z^{(2)}(k)\big), \qquad (3.16)$$

$$\big(\tau^{(1)}(k+1), \tau^{(2)}(k+1)\big) = \big(\tau^{(1)}(k), \tau^{(2)}(k)\big) \text{ with probability } 1 - a \cdot \eta \cdot s\big(Z^{(1)}(k), Z^{(2)}(k)\big).$$

Here $\{\xi^{(1)}(k)\}_{k \geq 0}$ and $\{\xi^{(2)}(k)\}_{k \geq 0}$ are two sequences of independent and identically distributed $d$-dimensional Gaussian random vectors, $\eta > 0$ is the stepsize, while $a$ and $s(\cdot, \cdot)$ are specified in (2.6). Note that $\eta$ and $a$ are chosen such that $\eta \cdot a < 1$ in order to define a valid probability.

Hereafter we denote by $\{Z^\eta(k)\}_{k \geq 0}$ the iterates of the algorithm specified in (3.15) and (3.16). We use the superscript $\eta$ to emphasize that the iterates depend on the stepsize $\eta$. Let $\{Z_t^\eta\}_{t \geq 0}$ be the continuous-time interpolation of $\{Z^\eta(k)\}_{k \geq 1}$, which is a continuous-time stochastic process defined as

$$Z_t^\eta = Z_0 - \int_0^t \nabla U\big(Z_{\lfloor s/\eta \rfloor \eta}^\eta\big)\mathrm{d}s + \int_0^t \Sigma_{\lfloor s/\eta \rfloor \eta}^\eta \mathrm{d}W_s, \qquad (3.17)$$

where $\Sigma_{k\eta}^\eta = \mathrm{diag}\{\sqrt{2\tau^{(1)}(k)} \cdot I_d, \sqrt{2\tau^{(2)}(k)} \cdot I_d\}$. Then for all integers $k \geq 0$ and $t = k\eta$, we have $Z_t^\eta = Z_{k\eta}^\eta = Z^\eta(k)$.

In the following theorem, we characterize the discretization error by the mean squared error between the continuous-time temperature swapping Langevin diffusion in (3.14) and the continuous-time interpolation in (3.17) of the discrete-time algorithm in (3.15) and (3.16). In specific, we consider a fixed time interval $[0, T]$ and upper bound the mean squared error $\mathbb{E}[\|Z_t - Z_t^\eta\|^2]$ for all $t \in [0, T]$. The following theorem shows that this error grows linearly with respect to the stepsize $\eta$.

**Theorem 3.6.** Under Assumption 3.1, there exists a constant $\gamma(d, \tau_1, \tau_2, a, L, \alpha, \beta, T)$ that only depends on the dimension $d$, temperature parameters $\tau_1$ and $\tau_2$, swapping intensity $a$, smoothness constant $L$ and dissipative constants $(\alpha, \beta)$ of $U(\cdot)$, and length of the time interval $T$, such that for all $t \in [0, T]$,

$$\mathbb{E}\big[\|Z_t - Z_t^\eta\|^2\big] \leq \gamma(d, \tau_1, \tau_2, a, L, \alpha, \beta, T) \cdot \eta,$$

provided that the stepsize $\eta$ satisfies $0 < \eta < \alpha/L^2$.

The proof of Theorem 3.6 adapts from the proof framework of Theorem 5.13 in Yin & Zhu (2010). We defer the detailed proof to §C of the appendix.

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

## A    DISCUSSION AND CONCLUSION

In this paper, we apply the idea of replica exchange to the Langevin diffusion in the context of nonconvex optimization. In particular, we quantify the benefits of replica exchange compared with the standard Langevin diffusion from two perspectives, that is, the convergence of the $\chi^2$-divergence and the LDP of the empirical measures. We show from both perspectives that replica exchange accelerates the convergence towards the invariant distribution. In the following, we discuss several related issues.

**Flat and sharp minima:** The Langevin diffusion yields an invariant distribution that concentrates around the global minima. In particular, as illustrated in Keskar et al. (2016), flat minima generally achieves better generalization than sharp ones. Meanwhile, as observed in Zhang et al. (2018), the invariant distribution of the Langevin diffusion biases more towards the flat minima than the sharp minima. Intuitively, this observation follows from locally integrating the density of the invariant distribution, which is proportional to $\exp\{-U(x)/\tau\}$. As specified in (2.3), the invariant distribution of the replica exchange Langevin diffusion takes the same form, and hence processes the same desired property of biasing towards the flat minima.

**Choice of swapping intensity:** Recall that in §3 we prove that swapping accelerates the exponential rate of convergence of the $\chi^2$-divergence and increases the LDP rate function of the empirical measures. In particular, a large swapping intensity $a$ leads to a stronger acceleration effect. In continuous time, the swapping intensity $a$ should be as large as possible. However, in discrete time, to ensure the existence of a valid swapping probability in (3.16), $a$ can only be as large as $1/\eta$, since by (2.6) we have $s(\cdot, \cdot) \leq 1$. This constraint characterizes the tradeoff between the acceleration effect and the discretization effort, since more fine-grained discretization with a smaller stepsize $\eta$ allows for a larger swapping intensity $a$. Moreover, intuitively speaking, more frequent swapping makes the particles more volatile, which lowers the accuracy of discretization.

**Choice of temperatures:** In the replica exchange Langevin diffusion, we use two particles driven by low and high temperatures to achieve "local exploitation" and "global exploration", respectively. There is a fundamental tradeoff in the choice of the temperatures, especially the high temperature. In specific, a larger $\tau_2$ results in faster global exploration. However, if $\tau_2$ is too large, by the second equation in (2.4), the gradient term $\nabla U(\cdot)$ becomes negligible compared with the white noise term $\sqrt{2\tau_2}\mathrm{d}W_t$. Consequently, the second particle "blindly" explores the whole domain $\mathbb{R}^d$ uniformly at random, without adapting the geometry of the objective function. As discussed in §2, most swaps happen only when better positions with smaller objective values are discovered by the second particle. Thus, if $\tau_2$ is too large, the global exploration is fast but ineffective.

## B    BACKGROUND

In this section, we provide the background of Markov process, which is necessary for the theoretical analysis. We analyze the properties of Markov processes from the viewpoint of Markov semigroup. A semigroup is a family of linear operators on Banach space $\mathcal{C}(\mathbb{R}^d)$, the space of bounded continuous functions on $\mathbb{R}^d$ equipped with the uniform norm. Formally, we have the following definition.

**Definition B.1** (Semigroup of Operators). A family $\{P_t\}_{t\geq 0}$ of linear operators on $\mathcal{C}(\mathbb{R}^d)$ is called a semigroup if and only if it satisfies the following conditions:

(1) For $t = 0$, we have $P_t = I$, which is the identical mapping on $\mathcal{C}(\mathbb{R}^d)$.

(2) The map $t \rightarrow P_t$ is continuous in the sense that, for all $f \in \mathcal{C}(\mathbb{R}^d)$, $t \rightarrow P_t(f)$ is a continuous map from $\mathbb{R}^+$ to $\mathcal{C}(\mathbb{R}^d)$.

(3) For all $f \in \mathcal{C}(\mathbb{R}^d)$ and $s, t > 0$, we have $P_{s+t}(f) = P_s(P_t(f))$.

Moreover, a semigroup of operators is Markov if and only if the following additional conditions are satisfied:

(4) For all $t > 0$, we have $P_t(\mathbb{1}) = \mathbb{1}$, where $\mathbb{1}$ is the constant function with value one.

(5) For all $f \geq 0$, we have $P_t(f) \geq 0$.

Recall that a Markov process on $\mathbb{R}^d$ is a stochastic process $\{X_t\}_{t \geq 0}$ satisfying

$$\mathbb{P}(X_t \in \Gamma \,|\, \mathscr{F}_s) = \mathbb{P}(X_t \in \Gamma \,|\, X_s) \tag{B.1}$$

for all measurable sets $\Gamma$ and times $s < t$. Here $\mathscr{F}_s = \sigma(X_u, u \leq s)$, which is the natural filtration of the Markov process $\{X_t\}_{t \geq 0}$. Given a Markov process $\{X_t\}_{t \geq 0}$, we define a family of operators $\{P_t\}_{t \geq 0}$ on $C(\mathbb{R}^d)$, the space of bounded continuous functions on $\mathbb{R}^d$, as follows

$$P_t\big(f(x)\big) = \mathbb{E}\big[f(X_t) \,|\, X_0 = x\big]. \tag{B.2}$$

According to the Markov property in (B.1), we have

$$P_{t+s}(f) = P_t(P_s(f)) \tag{B.3}$$

and hence, $\{P_t\}_{t \geq 0}$ indeed forms a semigroup. We call $\{P_t\}_{t \geq 0}$ the Markov semigroup associated with the Markov process $\{X_t\}_{t \geq 0}$.

Given a Markov semigroup, we can define its infinitesimal generator, a linear operator that describes the Markov semigroup's behavior for an infinitesimal $t$. Formally, we have the following definition. See, e.g., Revuz & Yor (2013) for more details.

**Definition B.2** (Infinitesimal Generator of Markov Semigroup). The infinitesimal generator $\mathscr{L}$ of a Markov semigroup $\{P_t\}_{t \geq 0}$ is defined by

$$\mathscr{L}(f) = \lim_{t \to 0} \frac{P_t(f) - f}{t}$$

for all $f \in \mathcal{D}(\mathscr{L})$. Here $\mathcal{D}(\mathscr{L})$ denotes the subset of $C(\mathbb{R}^d)$ such that the above limit exists.

Intuitively, the infinitesimal generator can be viewed as the derivative of the Markov semigroup at time $t = 0$. It uniquely determines the Markov semigroup according to the semigroup property (B.3). In the following, we define the invariant and reversible measure with respect to a Markov semigroup.

**Definition B.3** (Invariance and Reversibility). For a Markov semigroup $\{P_t\}_{t \geq 0}$ whose infinitesimal generator is $\mathscr{L}$, a probability measure $\pi$ is invariant with respect to $\{P_t\}_{t \geq 0}$ if and only if

$$\int \mathscr{L}(f) \mathrm{d}\pi = 0$$

for all $t \geq 0$ and $f \in C(\mathbb{R}^d)$. A probability measure $\pi$ is reversible with respect to $\{P_t\}_{t \geq 0}$ if and only if

$$\int f \mathscr{L}(g) \mathrm{d}\pi = \int g \mathscr{L}(f) \mathrm{d}\pi$$

for all $f, g \in \mathcal{D}(\mathscr{L})$. Note that reversibility implies invariance when we plug $g = \mathbb{1}$ into the definition of reversibility.

We remark that the standard definitions of invariance and reversibility are based on the Markov semigroup $\{P_t\}_{t \geq 0}$ instead of its infinitesimal generator $\mathscr{L}$, which is equivalent to the following definition. The definitions via infinitesimal generator simplify the proof of our lemmas and theorems.

**Definition B.4** (Equivalent Definitions of Invariance and Reversibility ). For a Markov semigroup $\{P_t\}_{t \geq 0}$, a probability measure $\pi$ is invariant with respect to $\{P_t\}_{t \geq 0}$ if and only if

$$\int P_t f \mathrm{d}\pi = \int f \mathrm{d}\pi$$

for every $t \geq 0$ and $f \in C(\mathbb{R}^d)$. Reversibility is another important concept in Markov semigroup theory. A probability measure $\mu$ is reversible for $\{P_t\}_{t \geq 0}$ if and only if

$$\int f P_t g \mathrm{d}\pi = \int g P_t f \mathrm{d}\pi$$

for all $t \geq 0$ and $f, g \in C(\mathbb{R}^d)$. Based on the Definition B.2, we can show that this definition is equivalent to Definition B.3.

In the definitions of Markov semigroup and associated concepts, we restrict the test function $f$ to $\mathcal{C}(\mathbb{R}^d)$, the space of bounded continuous functions. However, for $\pi$ being the invariant distribution with respect to $\{P_t\}_{t \geq 0}$, we can always (slightly) extend the definition of Markov semigroup to a larger function space $\bar{L}^2(\pi)$. We use $\{\bar{P}_t\}_{t \geq 0}$ and $\bar{\mathscr{L}}$ to denote the corresponding extended Markov semigroup and infinitesimal generator. One can show that, when $\pi$ is reversible, $\bar{\mathscr{L}}$ is a self-adjoint operator in the Hilbert space $L^2(\pi)$, which is equipped with the inner product $\langle f, g \rangle_\pi = \int f g \mathrm{d}\pi$. Furthermore, one can show that $\bar{\mathscr{L}}$ is positive semidefinite. In this paper, for the convenience of discussion, we do not distinguish the slight difference between $\mathscr{L}$ and its extension $\bar{\mathscr{L}}$, and also the difference of $\mathcal{D}(\mathscr{L})$, $\mathcal{C}(\mathbb{R}^d)$, and $L^2(\pi)$, since $\mathcal{C}(\mathbb{R}^d)$ is dense in $L^2(\pi)$ and $\mathcal{D}(\mathscr{L})$ is dense in $\mathcal{C}(\mathbb{R}^d)$ according to Hille-Yosida theorem (Phillips & Hille, 1957). When we choose a test function, we also assume that all the related operations are well-defined.

## C  DETAILED PROOFS

In this section, we present the detailed proofs for all the theorems and lemmas in this paper.

### C.1  PROOF OF LEMMA 3.2

*Proof.* The dynamics in (2.4)-(2.6) defines a Langevin diffusion with jump. Its generator takes the form

$$\mathscr{L}^a\big(f(x_1, x_2)\big) = \underbrace{-\big\langle \nabla_{x_1} f(x_1, x_2), \nabla_{x_1} U(x_1) \big\rangle + \tau_1 \Delta_{x_1} f(x_1, x_2)}_{\mathscr{L}_1^a\big(f(x_1, x_2)\big)}$$

$$\underbrace{-\big\langle \nabla_{x_2} f(x_1, x_2), \nabla_{x_2} U(x_2) \big\rangle + \tau_2 \Delta_{x_2} f(x_1, x_2)}_{\mathscr{L}_2^a\big(f(x_1, x_2)\big)} + \underbrace{a \cdot s(x_1, x_2) \cdot \big(f(x_2, x_1) - f(x_1, x_2)\big)}_{\mathscr{L}_s^a\big(f(x_1, x_2)\big)}$$

(C.1)

and its domain $\mathcal{D}(\mathscr{L}^a) = \mathcal{C}_c^2(\mathbb{R}^{2d})$, which is the space of all twice-differentiable functions with compact support. The first two terms $\mathscr{L}_1^a$ and $\mathscr{L}_2^a$ on the right-hand side of (C.1) correspond to the standard Langevin diffusion, while the last term $\mathscr{L}_s^a$ arises from swapping. To prove invariance and reversibility, by Definition B.3, we only need to show that

$$\int_{\mathbb{R}^d} \int_{\mathbb{R}^d} g(x_1, x_2) \mathscr{L}^a\big(f(x_1, x_2)\big) \mathrm{d}\pi(x_1, x_2)$$

$$= \int_{\mathbb{R}^d} \int_{\mathbb{R}^d} f(x_1, x_2) \mathscr{L}^a\big(g(x_1, x_2)\big) \mathrm{d}\pi(x_1, x_2)$$

(C.2)

for all $f, g \in \mathcal{D}(\mathscr{L}^a)$, where $\pi$ is defined in (3.4).

Note that for the Laplacian term in $\mathscr{L}_1^a$ on the right-hand side of (C.1), by integration by parts and the fact that $f$ and $g$ have compact support, we have that for all fixed $x_2 \in \mathbb{R}^d$,

$$- \int_{\mathbb{R}^d} g(x_1, x_2) \exp\big(-U(x_1)/\tau_1\big) \Delta_{x_1} f(x_1, x_2) \mathrm{d}x_1 \tag{C.3}$$

$$= \int_{\mathbb{R}^d} \Big\langle \nabla_{x_1} \big[g(x_1, x_2) \exp\big(-U(x_1)/\tau_1\big)\big], \nabla_{x_1} f(x_1, x_2) \Big\rangle \mathrm{d}x_1$$

$$= \int_{\mathbb{R}^d} \Big\langle \exp\big(-U(x_1)/\tau_1\big) \nabla_{x_1} g(x_1, x_2) - \exp\big(-U(x_1)/\tau_1\big) g(x_1, x_2) \nabla_{x_1} U(x_1), \nabla_{x_1} f(x_1, x_2) \Big\rangle \mathrm{d}x_1.$$

Recall that $\mu(x_1, x_2)$ is defined in (2.5). Hence, (C.3) takes the equivalent form

$$\int_{\mathbb{R}^d} g(x_1, x_2) \mathscr{L}_1^a\big(f(x_1, x_2)\big) \mu(x_1, x_2) \mathrm{d}x_1 = - \int_{\mathbb{R}^d} \big\langle \nabla_{x_1} f(x_1, x_2), \nabla_{x_1} g(x_1, x_2) \big\rangle \mu(x_1, x_2) \mathrm{d}x_1,$$

and hence,

$$\int_{\mathbb{R}^d} \int_{\mathbb{R}^d} g(x_1, x_2) \mathscr{L}_1^a\big(f(x_1, x_2)\big) \mu(x_1, x_2) \mathrm{d}x_1 \mathrm{d}x_2$$

$$= - \int_{\mathbb{R}^d} \int_{\mathbb{R}^d} \big\langle \nabla_{x_1} f(x_1, x_2), \nabla_{x_1} g(x_1, x_2) \big\rangle \mu(x_1, x_2) \mathrm{d}x_1 \mathrm{d}x_2. \tag{C.4}$$

By switching the positions of $f$ and $g$ in (C.4), we have

$$\int_{\mathbb{R}^d} \int_{\mathbb{R}^d} f(x_1, x_2) \mathscr{L}_1^a\big(g(x_1, x_2)\big) \mu(x_1, x_2) \mathrm{d}x_1 \mathrm{d}x_2$$

$$= - \int_{\mathbb{R}^d} \int_{\mathbb{R}^d} \big\langle \nabla_{x_1} g(x_1, x_2), \nabla_{x_1} f(x_1, x_2) \big\rangle \mu(x_1, x_2) \mathrm{d}x_1 \mathrm{d}x_2. \tag{C.5}$$

By (C.4) and (C.5) we have

$$\int_{\mathbb{R}^d} \int_{\mathbb{R}^d} g(x_1, x_2) \mathscr{L}_1^a\big(f(x_1, x_2)\big) \mu(x_1, x_2) \mathrm{d}x_1 \mathrm{d}x_2$$

$$= \int_{\mathbb{R}^d} \int_{\mathbb{R}^d} f(x_1, x_2) \mathscr{L}_1^a\big(g(x_1, x_2)\big) \mu(x_1, x_2) \mathrm{d}x_1 \mathrm{d}x_2. \tag{C.6}$$

By the same derivation, (C.6) also holds for $\mathscr{L}_2^a$. Thus, it remains to prove that (C.6) holds for $\mathscr{L}_s^a$ as well. For notational simplicity, in the following we use $f^+$ and $f^-$ to denote $f(x_1, x_2)$ and $f(x_2, x_1)$ respectively, and define $g^+$, $g^-$, $\mu^+$, and $\mu^-$ in a similar way. Then we have

$$\int_{\mathbb{R}^d} \int_{\mathbb{R}^d} g^+ \big(1 \wedge (\mu^-/\mu^+)\big)(f^- - f^+) \mu^+ \mathrm{d}x_1 \mathrm{d}x_2$$

$$= \int\int_{\{(x_1,x_2):\mu^- \leq \mu^+\}} g^+ \mu^-(f^- - f^+) \mathrm{d}x_1 \mathrm{d}x_2 + \int\int_{\{(x_1,x_2):\mu^- > \mu^+\}} g^+ \mu^+(f^- - f^+) \mathrm{d}x_1 \mathrm{d}x_2$$

$$= \int\int_{\{(x_1,x_2):\mu^- \leq \mu^+\}} g^+ \mu^-(f^- - f^+) \mathrm{d}x_1 \mathrm{d}x_2 + \int\int_{\{(x_1,x_2):\mu^+ > \mu^-\}} g^- \mu^-(f^+ - f^-) \mathrm{d}x_2 \mathrm{d}x_1$$

$$= \int\int_{\{(x_1,x_2):\mu^- \leq \mu^+\}} \mu^-(g^+ f^- + g^- f^+ - g^+ f^+ - g^- f^-) \mathrm{d}x_1 \mathrm{d}x_2, \tag{C.7}$$

where the third equality follows from symmetry. Similar to (C.7), we also have

$$\int_{\mathbb{R}^d} \int_{\mathbb{R}^d} f^+ \big(1 \wedge (\mu^-/\mu^+)\big)(g^- - g^+) \mu^+ \mathrm{d}x_1 \mathrm{d}x_2$$

$$= \int\int_{\{(x_1,x_2):\mu^- < \mu^+\}} \mu^-(f^+ g^- + f^- g^+ - f^+ g^+ - f^- g^-) \mathrm{d}x_1 \mathrm{d}x_2. \tag{C.8}$$

Combining (C.7) and (C.8), we obtain

$$\int_{\mathbb{R}^d} \int_{\mathbb{R}^d} g^+ \big(1 \wedge (\mu^-/\mu^+)\big)(f^- - f^+) \mu^+ \mathrm{d}x_1 \mathrm{d}x_2 = \int_{\mathbb{R}^d} \int_{\mathbb{R}^d} f^+ \big(1 \wedge (\mu^-/\mu^+)\big)(g^- - g^+) \mu^+ \mathrm{d}x_1 \mathrm{d}x_2,$$

which is equivalent to

$$\int_{\mathbb{R}^d}\int_{\mathbb{R}^d} g(x_1,x_2)\mathscr{L}_s^a\big(f(x_1,x_2)\big)\mu(x_1,x_2)\mathrm{d}x_1\mathrm{d}x_2 = \int_{\mathbb{R}^d}\int_{\mathbb{R}^d} f(x_1,x_2)\mathscr{L}_s^a\big(g(x_1,x_2)\big)\mu(x_1,x_2)\mathrm{d}x_1\mathrm{d}x_2$$

by the definition of $\mathscr{L}_s^a$ in (C.1). In summary, we obtain that (C.2) holds for $\pi$ defined in (3.4). Thus, $\{Z_t\}_{t\geq 0}$ is a reversible Markov process with the invariant distribution $\mathrm{d}\pi(x_1,x_2)\propto \exp(-U(x_1)/\tau_1 - U(x_2)/\tau_2)\mathrm{d}x_1\mathrm{d}x_2$. $\qquad\square$

### C.2 PROOF OF THEOREM 3.3

*Proof.* Recall the infinitesimal generator of the replica exchange Langevin diffusion with swapping intensity $a$ defined in (3.3). According to the properties of $\nabla$ and $\Delta$, for $i = 1, 2$, we have

$$\nabla_{x_i} f^2(x_1,x_2) = 2f(x_1,x_2)\cdot\nabla_{x_i}f(x_1,x_2)$$
$$\Delta_{x_i} f^2(x_1,x_2) = 2f(x_1,x_2)\cdot\Delta_{x_i}f(x_1,x_2) + 2\|\nabla_{x_i}f(x_1,x_2)\|^2.$$

Then by the definition of the Carré du Champ operator $\Gamma^a$ in (3.5), we have

$$\Gamma^a\big(f(x_1,x_2)\big) = 1/2\cdot\mathcal{L}^a\big(f(x_1,x_2)^2\big) - f(x_1,x_2)\cdot\mathcal{L}^a\big(f(x_1,x_2)\big)$$
$$= \tau_1\cdot\big\|\nabla_{x_1}f(x_1,x_2)\big\|^2 + \tau_2\cdot\big\|\nabla_{x_2}f(x_1,x_2)\big\|^2 + a/2\cdot s(x_1,x_2)\cdot\big(f(x_2,x_1) - f(x_1,x_2)\big)^2.$$

Hence, the Dirichlet form $\mathscr{E}^a$ is given by

$$\mathscr{E}^a(f) = \int\Big(\tau_1\cdot\big\|\nabla_{x_1}f(x_1,x_2)\big\|^2 + \tau_2\cdot\big\|\nabla_{x_2}f(x_1,x_2)\big\|^2\Big)\mathrm{d}\pi(x_1,x_2)$$
$$+ \int a/2\cdot s(x_1,x_2)\cdot\big(f(x_2,x_1) - f(x_1,x_2)\big)^2\mathrm{d}\pi(x_1,x_2),$$

which concludes the proof of Theorem 3.3. $\qquad\square$

### C.3 PROOF OF THEOREM 3.4

*Proof.* Note that the replica exchange Langevin diffusion without swapping, that is, $a = 0$, shares the same invariant distribution in (2.5) with the ones with swapping, that is, $a > 0$. Furthermore, Theorem 3.3 shows that $\mathscr{E}^a(f) \geq \mathscr{E}^0(f)$ for all $a > 0$. Hence it remains to show that the replica exchange Langevin diffusion without swapping satisfies the Poincaré inequality in (3.10).

Recall that the replica exchange Langevin diffusion without swapping is defined as

$$\mathrm{d}Z_t^{(1)} = -\nabla U(Z_t^{(1)})\mathrm{d}t + \sqrt{2\tau_1}\mathrm{d}W_t^{(1)}, \quad \mathrm{d}Z_t^{(2)} = -\nabla U(Z_t^{(2)})\mathrm{d}t + \sqrt{2\tau_2}\mathrm{d}W_t^{(2)}, \qquad (\text{C.9})$$

and its infinitesimal generator is given by

$$\mathscr{L}^0\big(f(x_1,x_2)\big) = -\big\langle\nabla_{x_1}f(x_1,x_2), \nabla_{x_1}U(x_1)\big\rangle + \tau_1\cdot\Delta_{x_1}f(x_1,x_2)$$
$$- \big\langle\nabla_{x_2}f(x_1,x_2), \nabla_{x_2}U(x_2)\big\rangle + \tau_2\cdot\Delta_{x_2}f(x_1,x_2). \qquad (\text{C.10})$$

The Poincaré inequality for the replica exchange Langevin diffusion in (C.9) is a direct consequence of the existence of a Lyapunov function. In the specific, we have the following lemma, which is adapted from Theorem 1.4 in Bakry et al. (2008).

**Lemma C.1** (Lyapunov implies Poincaré). Let $\{Z_t\}_{t\geq 0}$ be the replica exchange Langevin diffusion without swapping on $\mathbb{R}^{2d}$, whose infinitesimal generator is $\mathscr{L}^0$ defined in (C.10). We further assume that there exists a function $V(x_1,x_2)\geq 1$, and constants $\lambda > 0$, $b\geq 0$, and $r > 0$ such that

$$\mathscr{L}^0\big(V(x_1,x_2)\big)\leq -\lambda\cdot V(x_1,x_2) + b\cdot\mathbb{1}_{\mathcal{B}_r}(x_1,x_2), \qquad (\text{C.11})$$

where $\mathbb{1}_{\mathcal{B}_r}$ denotes the indicator function of the centered ball with radius $r$ in $\mathbb{R}^{2d}$. Then $\{Z_t\}_{t\geq 0}$ satisfies the Poincaré inequality. We call $V(x_1,x_2)$ the Lyapunov function.

*Proof.* We extend the proof of Theorem 1.4 in Bakry et al. (2008), which characterizes the standard Langevin diffusion with a single temperature, to the setting with two temperatures. For completeness, we provide a detailed proof in §C.4. □

When the loss function $U(x)$ satisfies the $(\alpha, \beta)$-dissipative condition in Assumption 3.1, following Raginsky et al. (2017), we construct the following Lyapunov function

$$V(x_1, x_2) = \exp\left\{\alpha/4 \cdot \left(\|x_1\|^2/\tau_1 + \|x_2\|^2/\tau_2\right)\right\}.$$

A direct calculation shows that $V(x_1, x_2)$ satisfies the condition in (C.11). Hence, we apply Lemma C.1 to obtain that the replica exchange Langevin diffusion without swapping satisfies the Poincaré inequality. In summary, we prove that the Poincaré inequality holds for the replica exchange Langevin diffusion, which concludes the proof. □

## C.4 PROOF OF LEMMA C.1

*Proof.* We first prove the Poincaré inequality in a more general form. For all functions $f(x_1, x_2)$ : $\mathbb{R}^{2d} \to \mathbb{R}$, we show that if there exists a function $V(x_1, x_2) \geq 1$ such that (C.11) holds, then there exists a constant $\rho$ such that

$$\mathrm{Var}_\pi(f) \leq \rho \cdot \int \left(\tau_1 \cdot \left\|\nabla_{x_1} f(x_1, x_2)\right\|^2 + \tau_2 \cdot \left\|\nabla_{x_2} f(x_1, x_2)\right\|^2\right) \cdot \mathrm{d}\pi(x_1, x_2),$$

where $\tau_1$ and $\tau_2$ are the temperatures, and $\pi(x_1, x_2)$ is the invariant distribution. By setting the function $f$ as the Radon-Nykodim derivative $\mathrm{d}\nu/\mathrm{d}\pi$, we obtain the Poincaré inequality for the $\chi^2$-divergence in (3.10). Note that for all constants $u \in \mathbb{R}$, we have

$$\mathrm{Var}_\pi(f) \leq \int \left(f(x_1, x_2) - u\right)^2 \cdot \mathrm{d}\pi(x_1, x_2),$$

where $\mathrm{Var}_\pi(f)$ denotes the variance of $f(X_1, X_2)$ when $(X_1, X_2)$ follows the invariant distribution $\pi$. We use $f_u$ to denote the function $f - u$. By multiplying $f_u^2(x_1, x_2)/(\lambda V(x_1, x_2))$ on the both sides of (C.11) and integrating them over $\pi(x_1, x_2)$, we have

$$\int f_u(x_1, x_2)^2 \cdot \mathrm{d}\pi(x_1, x_2) \leq \int -\mathscr{L}^0\left(V(x_1, x_2)\right)/\left(\lambda V(x_1, x_2)\right) \cdot f_u(x_1, x_2)^2 \cdot \mathrm{d}\pi(x_1, x_2)$$
$$+ \int b \cdot \mathbb{1}_{\mathcal{B}_r}(x_1, x_2)/\left(\lambda V(x_1, x_2)\right) \cdot f_u(x_1, x_2)^2 \cdot \mathrm{d}\pi(x_1, x_2).$$
$$\text{(C.12)}$$

In the sequel, we upper bound the two terms on the right-hand side of (C.12).

For the first term on the right-hand side of (C.12), we invoke the divergence theorem, which states that for a scalar-valued function $g$, whose value at infinity is zero, and a vector-valued function $h$, for the integration under the measure $\omega$ we have

$$\int \left(\langle \nabla g, h \rangle + g \cdot \mathrm{div}\, h\right) \mathrm{d}\omega = \int \mathrm{div}(g \cdot h) \mathrm{d}\omega = 0. \tag{C.13}$$

In (C.13), by setting $\mathrm{d}\omega$ as the Lebesgue measure $\mathrm{d}x_1 \mathrm{d}x_2$ and

$$g = \exp\left(-U(x_1)/\tau_1 - U(x_2)/\tau_2\right) \quad \text{and} \quad h = f_u(x_1, x_2)^2/\left(\lambda V(x_1, x_2)\right) \cdot \nabla V(x_1, x_2),$$

we have

$$\int -\mathscr{L}^0\left(V(x_1, x_2)\right)/\left(\lambda V(x_1, x_2)\right) \cdot f_u(x_1, x_2)^2 \cdot \mathrm{d}\pi(x_1, x_2)$$
$$= \tau_1/\lambda \cdot \int \left\langle \nabla_{x_1}\left[f_u(x_1, x_2)^2/V(x_1, x_2)\right], \nabla_{x_1} V(x_1, x_2)\right\rangle \cdot \mathrm{d}\pi(x_1, x_2)$$
$$+ \tau_2/\lambda \cdot \int \left\langle \nabla_{x_2}\left[f_u(x_1, x_2)^2/V(x_1, x_2)\right], \nabla_{x_2} V(x_1, x_2)\right\rangle \cdot \mathrm{d}\pi(x_1, x_2). \tag{C.14}$$

Meanwhile, for the first term on the right-hand side of (C.14) we have

$$\int \Big\langle \nabla_{x_1}\big[f_u^2(x_1, x_2)/V(x_1, x_2)\big], \nabla_{x_1}V(x_1, x_2)\Big\rangle \cdot \mathrm{d}\pi(x_1, x_2)$$

$$= \int \Big(\big\|\nabla_{x_1}f_u(x_1, x_2)\big\|^2 - \big\|\nabla_{x_1}f_u(x_1, x_2) - \big(f_u(x_1, x_2)/V(x_1, x_2)\big) \cdot \nabla_{x_1}V(x_1, x_2)\big\|^2\Big) \cdot \mathrm{d}\pi(x_1, x_2)$$

$$\leq \int \big\|\nabla_{x_1}f_u(x_1, x_2)\big\|^2 \cdot \mathrm{d}\pi(x_1, x_2), \tag{C.15}$$

where the first equality follows from expanding the gradient of $f_u^2(x_1, x_2)/V(x_1, x_2)$ with respect to $x_1$. By a similar derivation, the same inequality holds for the second term on the right-hand side of (C.14) as well. Hence, plugging them into (C.15), for the first term on the right-hand side of (C.12) we have

$$\int -\mathscr{L}^0\big(V(x_1, x_2)\big)/\big(\lambda V(x_1, x_2)\big) \cdot f_u(x_1, x_2)^2 \cdot \mathrm{d}\pi(x_1, x_2)$$

$$\leq 1/\lambda \cdot \int \Big(\tau_1\big\|\nabla_{x_1}f_u(x_1, x_2)\big\|^2 + \tau_2\big\|\nabla_{x_2}f_u(x_1, x_2)\big\|^2\Big) \cdot \mathrm{d}\pi(x_1, x_2). \tag{C.16}$$

For the second term on the right-hand side of (C.12), we restrict our attention to the bounded domain $\mathcal{B}_r$. Based on the assumption that $V(x_1, x_2) \geq 1$, there exists a positive constant $k$ such that

$$\int b \cdot \mathbb{1}_{\mathcal{B}_r}(x_1, x_2)/\big(\lambda V(x_1, x_2)\big) \cdot f_u(x_1, x_2)^2 \cdot \mathrm{d}\pi(x_1, x_2)$$

$$\leq b/\lambda \cdot \int_{\mathcal{B}_r} f_u(x_1, x_2)^2 \cdot \mathrm{d}\pi(x_1, x_2)$$

$$\leq k \cdot \int_{\mathcal{B}_r} \big\|\nabla f_u(x_1, x_2)\big\|^2 \cdot \mathrm{d}\pi(x_1, x_2) + \left(\int_{\mathcal{B}_r} f_u(x_1, x_2) \cdot \mathrm{d}\pi(x_1, x_2)\right)^2 \Big/ \pi\big(\mathcal{B}_r\big), \tag{C.17}$$

where the second inequality follows from the Poincaré inequality for measures on a bounded domain (Bakry et al., 2008). Since (C.17) holds for all functions $f_u(x_1, x_2)$, one can choose a suitable $u^*$ such that $\int_{\mathcal{B}_r} f_{u^*}\mathrm{d}\pi = 0$. Then we have

$$\int b \cdot \mathbb{1}_{\mathcal{B}_r}(x_1, x_2)/\big(\lambda V(x_1, x_2)\big) \cdot f_u(x_1, x_2)^2 \cdot \mathrm{d}\pi(x_1, x_2)$$

$$\leq k \cdot \int_{\mathcal{B}_r} \big\|\nabla f_{u^*}(x_1, x_2)\big\|^2 \cdot \mathrm{d}\pi(x_1, x_2). \tag{C.18}$$

Finally, by plugging (C.16) and (C.18) into (C.12), we have that there exists a positive constant $\rho$ such that

$$\mathrm{Var}_\pi(f) \leq \int f_{u^*}(x_1, x_2)^2 \cdot \mathrm{d}\pi(x_1, x_2)$$

$$\leq \rho \cdot \int \Big(\tau_1\big\|\nabla_{x_1}f(x_1, x_2)\big\|^2 + \tau_2\big\|\nabla_{x_2}f(x_1, x_2)\big\|^2\Big) \cdot \mathrm{d}\pi(x_1, x_2),$$

which concludes the proof of Lemma C.1. □

### C.5 Proof of Theorem 3.5

According to Lemma 3.2, the replica exchange Langevin diffusion $\{Z_t\}_{t\geq 0}$ is reversible. Hence, we can apply the Donsker-Varadhan theory, which states that for a continuous-time reversible Markov process with invariant distribution $\pi$ and infinitesimal generator $\mathscr{L}$, LDP holds and its LDP rate function takes the explicit form

$$I(\nu) = \begin{cases} \big\|\sqrt{-\mathscr{L}}\big(\sqrt{d\nu/d\pi}\big)\big\|_\pi^2, & \text{if } \nu \ll \pi, \\ \infty, & \text{otherwise.} \end{cases} \tag{C.19}$$

Here $\sqrt{\mathscr{L}}$ denotes the square root of $-\mathscr{L}$, which is defined as follows.

Let $A$ be a positive semidefinite self-adjoint operator in the Hilbert space $(\mathcal{H}, \langle \cdot, \cdot \rangle_{\mathcal{H}})$, the square root of $A$ is defined as the self-adjoint operator $B$ such that $A = B^2$, which means $\langle x, Ax \rangle_{\mathcal{H}} = \langle x, B^2 x \rangle_{\mathcal{H}} = \langle Bx, Bx \rangle_{\mathcal{H}}$ for every $x \in \mathcal{H}$. As explained in §B, the infinitesimal generator $-\mathscr{L}$ can be extended to a positive semidefinite self-adjoint operator in the Hilbert space $L^2(\pi)$. We ignore the slight differences caused by extension. Hence, $\sqrt{-\mathscr{L}}$ is well-defined here.

Recall that the infinitesimal generator $\mathscr{L}^a$ of the replica exchange Langevin diffusion is given by (3.3). We first derive the explicit form of $\|\sqrt{-\mathscr{L}^a}(f)\|_\pi^2$. Since the square root of a positive semidefinite self-adjoint operator is self-adjoint, we have

$$\left\| \sqrt{-\mathscr{L}^a}(f) \right\|_\pi^2 = \left\langle \sqrt{-\mathscr{L}^a}(f), \sqrt{-\mathscr{L}^a}(f) \right\rangle_\pi = \left\langle f, \sqrt{-\mathscr{L}^a}^2(f) \right\rangle_\pi = \left\langle f, -\mathscr{L}^a(f) \right\rangle_\pi = -\int f \mathscr{L}^a(f) \mathrm{d}\pi.$$

Since $\pi$ is the invariant distribution, by Definition B.3 we have $\int \mathscr{L}^a(f^2) \mathrm{d}\pi = 0$. Then we have

$$\left\| \sqrt{-\mathscr{L}^a}(f) \right\|_\pi^2 = 1/2 \cdot \int \left( \mathscr{L}^a(f^2) - 2f\mathscr{L}^a(f) \right) \mathrm{d}\pi,$$

which is exactly the Dirichlet form. Then based on (3.3), we obtain

$$\mathscr{L}^a(f^2) - 2f\mathscr{L}^a(f) = 2\tau_1 \cdot \left\| \nabla_{x_1} f(x_1, x_2) \right\|^2 + 2\tau_2 \cdot \left\| \nabla_{x_2} f(x_1, x_2) \right\|^2$$
$$+ a \cdot s(x_1, x_2) \cdot \left( f(x_2, x_1) - f(x_1, x_2) \right)^2.$$

Hence, we have that for probability measures $\nu \ll \pi$,

$$I^a(\nu) = \int \tau_1 \left\| \nabla_{x_1} \sqrt{\mathrm{d}\nu/\mathrm{d}\pi(x_1, x_2)} \right\|^2 + \tau_2 \left\| \nabla_{x_2} \sqrt{\mathrm{d}\nu/\mathrm{d}\pi(x_1, x_2)} \right\|^2$$
$$+ a/2 \cdot s(x_1, x_2) \cdot \left( \sqrt{\mathrm{d}\nu/\mathrm{d}\pi(x_2, x_1)} - \sqrt{\mathrm{d}\nu/\mathrm{d}\pi(x_1, x_2)} \right)^2 \mathrm{d}\pi, \tag{C.20}$$

and $I(\nu) = \infty$, otherwise.

## C.6 Proof of Theorem 3.6

*Proof.* Recall that the temperature swapping Langevin diffusion $\{Z_t\}_{t \geq 0}$ and the continuous-time interpolated process $\{Z_t^\eta\}_{t \geq 0}$ are defined in (3.14) and (3.17). For all $t \in [0, T]$, we have

$$Z_t - Z_t^\eta = -\int_0^t \left( \nabla U(Z_s) - \nabla U\left( Z_{\lfloor s/\eta \rfloor \eta}^\eta \right) \right) \mathrm{d}s + \int_0^t \left( \Sigma_s - \Sigma_{\lfloor s/\eta \rfloor \eta}^\eta \right) \mathrm{d}W_s.$$

By applying the Cauchy-Schwartz inequality and then taking expectation, we have

$$\mathbb{E}\left[ \|Z_t - Z_t^\eta\|^2 \right] \leq 2 \cdot \mathbb{E}\left[ \left\| \int_0^t \left( \nabla U(Z_s) - \nabla U\left( Z_{\lfloor s/\eta \rfloor \eta}^\eta \right) \right) \mathrm{d}s \right\|^2 \right]$$
$$+ 2 \cdot \mathbb{E}\left[ \left\| \int_0^t \left( \Sigma_s - \Sigma_{\lfloor s/\eta \rfloor \eta}^\eta \right) \mathrm{d}W_s \right\|^2 \right]. \tag{C.21}$$

In the sequel, we upper bound the two terms on the right-hand side of (C.21).

For the first term, by Cauchy-Schwarz inequality, we have

$$\mathbb{E}\left[ \left\| \int_0^t \left( \nabla U(Z_s) - \nabla U\left( Z_{\lfloor s/\eta \rfloor \eta}^\eta \right) \right) \mathrm{d}s \right\|^2 \right]$$
$$\leq t \cdot \mathbb{E}\left[ \int_0^t \left\| \nabla U(Z_s) - \nabla U\left( Z_{\lfloor s/\eta \rfloor \eta}^\eta \right) \right\|^2 \mathrm{d}s \right]$$
$$\leq 2L^2 t \cdot \left( \mathbb{E}\left[ \int_0^t \|Z_s - Z_s^\eta\|^2 \mathrm{d}s \right] + \mathbb{E}\left[ \int_0^t \|Z_s^\eta - Z_{\lfloor s/\eta \rfloor \eta}^\eta\|^2 \mathrm{d}s \right] \right), \tag{C.22}$$

where the second inequality follows from the $L$-smoothness of $U(\cdot)$ in Assumption 3.1. In the following, we upper bound the second term on the right-hand side of (C.22). Note that we have

$$\mathbb{E}\left[\int_0^t \left\|Z_s^\eta - Z_{\lfloor s/\eta\rfloor\eta}^\eta\right\|^2 \mathrm{d}s\right] \leq \sum_{k=0}^{\lfloor t/\eta\rfloor} \mathbb{E}\left[\int_{k\eta}^{(k+1)\eta} \left\|Z_s^\eta - Z_{\lfloor s/\eta\rfloor\eta}^\eta\right\|^2 \mathrm{d}s\right]. \tag{C.23}$$

For all integers $k \geq 0$ and $s \in [k\eta, (k+1)\eta)$, based on the definition of $\{Z_t^\eta\}_{t\geq 0}$ in (3.17), we have

$$\left\|Z_s^\eta - Z_{\lfloor s/\eta\rfloor\eta}^\eta\right\|^2 = \|Z_s^\eta - Z_{k\eta}^\eta\|^2 = \left\|-\nabla U(Z_{k\eta}^\eta)\cdot(s-k\eta) + \Sigma_{k\eta}^\eta \int_{k\eta}^s \mathrm{d}W_u\right\|^2.$$

By applying the Cauchy-Schwartz inequality again, we obtain

$$\left\|Z_s^\eta - Z_{\lfloor s/\eta\rfloor\eta}^\eta\right\|^2 \leq 2\cdot\left\|\nabla U(Z_{k\eta}^\eta)\right\|^2\cdot(s-k\eta)^2 + 2\cdot\left\|\Sigma_{k\eta}^\eta\int_{k\eta}^s \mathrm{d}W_u\right\|^2$$

$$= 2\cdot\left\|\nabla U(Z_{k\eta}^\eta) - \nabla U(x^*)\right\|^2\cdot(s-k\eta)^2 + 2\cdot\left\|\Sigma_{k\eta}^\eta\int_{k\eta}^s \mathrm{d}W_u\right\|^2, \tag{C.24}$$

where $x^*$ is a stationary point of $U(\cdot)$, and by definition $\nabla U(x^*) = 0$. Recall that by Assumption 3.1, $U(\cdot)$ is $L$-smooth. Based on (C.24), we have

$$\left\|Z_s^\eta - Z_{\lfloor s/\eta\rfloor\eta}^\eta\right\|^2 \leq 2L^2\cdot\|Z_{k\eta}^\eta - x^*\|^2\cdot(s-k\eta)^2 + 2\cdot\left\|\Sigma_{k\eta}^\eta\int_{k\eta}^s \mathrm{d}W_u\right\|^2$$

$$\leq 4L^2\cdot\left(\|Z_{k\eta}^\eta\|^2 + \|x^*\|^2\right)\cdot(s-k\eta)^2 + 2\cdot\left\|\Sigma_{k\eta}^\eta\int_{k\eta}^s \mathrm{d}W_u\right\|^2. \tag{C.25}$$

By integrating the second inequality in (C.25) over the interval $[k\eta, (k+1)\eta]$ and taking expectation, then plugging it into the right-hand side of (C.23), we obtain

$$\mathbb{E}\left[\int_{k\eta}^{(k+1)\eta}\left\|Z_s^\eta - Z_{\lfloor s/\eta\rfloor\eta}^\eta\right\|^2 \mathrm{d}s\right]$$

$$\leq 4L^2\cdot\left(\sup_{k\geq 0}\mathbb{E}\left[\|Z_{k\eta}^\eta\|^2\right] + \|x^*\|^2\right)\cdot\eta^3/3 + 2\cdot\int_{k\eta}^{(k+1)\eta}\mathbb{E}\left[\left\|\Sigma_{k\eta}^\eta\int_{k\eta}^s \mathrm{d}W_u\right\|^2\right]\mathrm{d}s. \tag{C.26}$$

Note that on the right-hand side of (C.26), $\Sigma_{k\eta}^\eta$ is a diagonal matrix with diagonal entries $\sqrt{2\tau_1}$ or $\sqrt{2\tau_2}$, then by Itô isometry, we have

$$\mathbb{E}\left[\left\|\Sigma_{k\eta}^\eta\int_{k\eta}^s \mathrm{d}W_u\right\|^2\right] = \sum_{j=1}^{2d}\mathbb{E}\left[\left(\Sigma_{k\eta}^\eta(j)\int_{k\eta}^s \mathrm{d}W_u^{(j)}\right)^2\right] \leq 4d\tau_2(s-k\eta),$$

where $\Sigma_{k\eta}^\eta(j)$ denotes the $j$-th diagonal entry of the matrix $\Sigma_{k\eta}^\eta$, and $W_u^{(j)}$ is the $j$-th component of the $2d$-dimensional Brownian motion $W_u$. Hence, we have

$$\int_{k\eta}^{(k+1)\eta}\mathbb{E}\left[\left\|\Sigma_{k\eta}^\eta\int_{k\eta}^s \mathrm{d}W_u\right\|^2\right]\mathrm{d}s \leq \int_{k\eta}^{(k+1)\eta} 4d\tau_2(s-k\eta)\mathrm{d}s = 2d\tau_2\eta^2, \tag{C.27}$$

which provides an upper bound for the right-hand side of (C.26). Then by plugging (C.27) into (C.26), we obtain

$$\mathbb{E}\left[\int_{k\eta}^{(k+1)\eta}\left\|Z_s^\eta - Z_{\lfloor s/\eta\rfloor\eta}^\eta\right\|^2 \mathrm{d}s\right] \leq 4L^2\cdot\left(\sup_{k\geq 0}\mathbb{E}\left[\|Z_{k\eta}^\eta\|^2\right] + \|x^*\|^2\right)\cdot\eta^3 + 4d\tau_2\eta^2. \tag{C.28}$$

Then by plugging (C.28) into (C.23), we have

$$\mathbb{E}\left[\int_0^t\left\|Z_s^\eta - Z_{\lfloor s/\eta\rfloor\eta}^\eta\right\|^2 \mathrm{d}s\right] \leq (1+t/\eta)\cdot\left(4L^2\cdot\left(\sup_{k\geq 0}\mathbb{E}\left[\|Z_{k\eta}^\eta\|^2\right] + \|x^*\|^2\right)\cdot\eta^3 + 4d\tau_2\eta^2\right). \tag{C.29}$$

The following lemma shows that if the discretization stepsize $\eta$ falls into the interval $(0, \alpha/L^2)$, $\{Z_{k\eta}^\eta\}_{k\geq 0}$ are uniformly upper bounded in the $L^2$ sense.

**Lemma C.2.** If $0 < \eta < \alpha/L^2$, there exists a constant $\delta_1(d, \tau_2, L, \alpha, \beta)$, which depends on the dimension $d$, the temperature parameter $\tau_2$, and the smoothness constant $L$ and dissipative constants $(\alpha, \beta)$ of $U(\cdot)$, such that

$$\sup_{k \geq 0} \mathbb{E}\big[\|Z_{k\eta}^\eta\|^2\big] \leq \delta_1(d, \tau_2, L, \alpha, \beta).$$

*Proof.* The proof idea is to show that the sequence $\{\mathbb{E}[\|Z_{k\eta}^\eta\|^2]\}_{k \geq 0}$ satisfies a contractive inequality based on the discretization scheme defined in (3.15). See §C.7 for a detailed proof. □

By applying Lemma C.2 to (C.29), we have that there exists a constant $\delta_2(d, \tau_2, L, \alpha, \beta)$ such that

$$\mathbb{E}\bigg[\int_0^t \big\|Z_s^\eta - Z_{\lfloor s/\eta \rfloor \eta}^\eta\big\|^2 \mathrm{d}s\bigg] \leq \delta_2(d, \tau_2, L, \alpha, \beta) \cdot \eta.$$

Then based on (C.22), we obtain the following inequality

$$\mathbb{E}\bigg[\bigg\|\int_0^t \big(\nabla U(Z_s) - \nabla U\big(Z_{\lfloor s/\eta \rfloor \eta}^\eta\big)\big)\mathrm{d}s\bigg\|^2\bigg]$$

$$\leq 2L^2 t \cdot \bigg(\mathbb{E}\bigg[\int_0^t \|Z_s - Z_s^\eta\|^2 \mathrm{d}s\bigg] + \delta_2(d, \tau_2, L, \alpha, \beta) \cdot \eta\bigg), \tag{C.30}$$

which establishes an upper bound for the first term on the right-hand side of (C.21).

It remains to upper bound the second term on the right-hand side of (C.21). According to Itô isometry, we have

$$\mathbb{E}\bigg[\bigg\|\int_0^t \big(\Sigma_s - \Sigma_{\lfloor s/\eta \rfloor \eta}^\eta\big)\mathrm{d}W_s\bigg\|^2\bigg] = \sum_{j=1}^{2d} \int_0^t \mathbb{E}\Big[\big(\Sigma_s(j) - \Sigma_{\lfloor s/\eta \rfloor \eta}^\eta(j)\big)^2\Big]\mathrm{d}s$$

$$\leq \sum_{j=1}^{2d} \sum_{k=0}^{\lceil t/\eta \rceil} \int_{k\eta}^{(k+1)\eta} \mathbb{E}\Big[\big(\Sigma_s(j) - \Sigma_{k\eta}^\eta(j)\big)^2\Big]\mathrm{d}s, \tag{C.31}$$

where $\Sigma_s(j)$ and $\Sigma_{k\eta}^\eta(j)$ are the $j$-th diagonal entries of $\Sigma_s$ and $\Sigma_{k\eta}^\eta$, respectively. Recall that $\Sigma_s$ and $\Sigma_s^\eta$ are diagonal matrices with all diagonal entires being $\sqrt{2\tau_1}$ or $\sqrt{2\tau_2}$. Then for all $j$, $k$, and possible realizations of $\Sigma_s(j)$ and $\Sigma_{k\eta}^\eta(j)$, we have

$$\int_{k\eta}^{(k+1)\eta} \mathbb{E}\Big[\big(\Sigma_s(j) - \Sigma_{k\eta}^\eta(j)\big)^2\Big]\mathrm{d}s = 4(\sqrt{\tau_2} - \sqrt{\tau_2})^2 \cdot \int_{k\eta}^{(k+1)\eta} \mathbb{P}\big(\Sigma_s(j) \neq \Sigma_{k\eta}^\eta(j)\big)\mathrm{d}s. \tag{C.32}$$

To upper bound the probability on the right-hand side of (C.32), we take the expectation conditioning $Z_k^\eta$, which yields

$$\int_{k\eta}^{(k+1)\eta} \mathbb{P}\big(\Sigma_s(j) \neq \Sigma_{k\eta}^\eta(j)\big)\mathrm{d}s = \mathbb{E}\bigg[\int_{k\eta}^{(k+1)\eta} \mathbb{P}\big(\Sigma_s(j) \neq \Sigma_{k\eta}^\eta(j) \,\big|\, Z_{k\eta}^\eta\big)\mathrm{d}s\bigg]. \tag{C.33}$$

Recall that the rate of swapping is specified in (2.6) and (2.7). The conditional probability on the right-hand side of (C.33) satisfies

$$\mathbb{P}\big(\Sigma_s(j) \neq \Sigma_{k\eta}^\eta(j) \,\big|\, Z_{k\eta}^\eta\big) = a \cdot s\big(Z_{k\eta}^{\eta(1)}, Z_{k\eta}^{\eta(2)}\big) \cdot (s - k\eta) + o(s - k\eta), \tag{C.34}$$

where $Z_{k\eta}^{\eta(1)}, Z_{k\eta}^{\eta(2)}$ are the first and second components of $Z_{k\eta}^\eta$, $s(\cdot, \cdot)$ is defined in (2.6), and $o(\cdot)$ is the little-$o$ notation, which denotes the higher-order term with respect to $s - k\eta$. Hence, by combining (C.32)-(C.34), we have that there exists a constant $\delta_3(\tau_1, \tau_2, a)$ such that

$$\int_{k\eta}^{(k+1)\eta} \mathbb{E}\Big[\big(\Sigma_s(j) - \Sigma_{k\eta}^\eta(j)\big)^2\Big]\mathrm{d}s = 4(\sqrt{\tau_2} - \sqrt{\tau_2})^2 \cdot \int_{k\eta}^{(k+1)\eta} \big(a \cdot (s - k\eta) + o(s - k\eta)\big)\mathrm{d}s$$

$$\leq \delta_3(\tau_1, \tau_2, a) \cdot \eta^2.$$

Then based on (C.31), there exists a constant $\delta_4(d, \tau_1, \tau_2, a)$ such that

$$\mathbb{E}\left[\left\|\int_0^t \left(\Sigma_s - \Sigma_{\lfloor s/\eta \rfloor \eta}^\eta\right) \mathrm{d}W_s\right\|^2\right] \leq 2d \cdot \left(\lceil t/\eta \rceil + 1\right) \cdot \delta_3(\tau_1, \tau_2, a) \cdot \eta^2 \leq \delta_4(d, \tau_1, \tau_2, a) \cdot t \cdot \eta. \tag{C.35}$$

Finally, by plugging (C.30) and (C.35) into (C.21), we have

$$\mathbb{E}\left[\|Z_t - Z_t^\eta\|^2\right] \leq 4L^2 t \cdot \left(\int_0^t \mathbb{E}\left[\|Z_s - Z_s^\eta\|^2\right]\mathrm{d}s + \delta_2(d, \tau_2, L, \alpha, \beta) \cdot \eta\right) + 2 \cdot \delta_4(d, \tau_1, \tau_2, a) \cdot t \cdot \eta.$$

Hence, by applying the Grönwall's inequality (Dragomir, 2003), we have that there exists a constant $\gamma(d, \tau_1, \tau_2, a, L, \alpha, \beta, T)$ such that for all $t \in [0, T]$,

$$\mathbb{E}\left[\|Z_t - Z_t^\eta\|^2\right] \leq \gamma(d, \tau_1, \tau_2, a, L, \alpha, \beta, T) \cdot \eta.$$

In other words, the mean squared error of discretization grows linearly with respect to the stepsize $\eta$, which concludes the proof of Theorem 3.6. $\square$

## C.7   PROOF OF LEMMA C.2

*Proof.* Recall that for all integers $k \geq 0$, based on the discretization scheme in (3.15) and (3.16), for $i \in \{1, 2\}$, we have

$$Z^{(i)}(k+1) = Z^{(i)}(k) - \eta \cdot \nabla U\left(Z^{(i)}(k)\right) + \sqrt{2\eta \cdot \tau^{(i)}(k)} \cdot \xi^{(i)}(k),$$

where $\xi^{(i)}(k)$ is a standard $d$-dimensional Gaussian random vector and the temperature $\tau^{(i)}(k)$ takes value in $\{\tau_1, \tau_2\}$. Also note that by the definition of the continuous-time interpolated process $\{Z_t^\eta\}_{t \geq 0}$ in (3.17), we have $Z(k) = Z_{k\eta}^\eta$. Hence, we have

$$\mathbb{E}\left[\|Z_{(k+1)\eta}^\eta\|^2\right] = \mathbb{E}\left[\|Z_{k\eta}^\eta - \eta \cdot \nabla U(Z_{k\eta}^\eta)\|^2\right] + \mathbb{E}\left[2\eta \cdot \tau^{(1)}(k) \cdot \|\xi^{(1)}(k)\|^2\right] + \mathbb{E}\left[2\eta \cdot \tau^{(2)}(k) \cdot \|\xi^{(2)}(k)\|^2\right]$$
$$+ 2 \cdot \mathbb{E}\left[\left\langle Z_{(k+1)\eta}^\eta - \eta \cdot \nabla U(Z_{k\eta}^\eta), \sqrt{2\eta \cdot \Sigma_{k\eta}^\eta} \cdot \xi(k)\right\rangle\right]. \tag{C.36}$$

Since $\xi(k)$ is independent of $Z_{k\eta}^\eta$ and by (3.16) the distribution of $\tau(k)$ is only determined by $Z_{k\eta}^\eta$, the last term on the right-hand side of (C.36) is zero. Moreover, note that each component of $\tau(k)$ only takes value in $\{\tau_1, \tau_2\}$ and $\tau_1 < \tau_2$. Then by (C.36) we have

$$\mathbb{E}\left[\|Z_{(k+1)\eta}^\eta\|^2\right] \leq \mathbb{E}\left[\|Z_{k\eta}^\eta - \eta \cdot \nabla U(Z_{k\eta}^\eta)\|^2\right] + 4d\eta\tau_2. \tag{C.37}$$

Recall that $U(\cdot)$ is $(\alpha, \beta)$-dissipative and $L$-smooth by Assumption 3.1. Then we have

$$\mathbb{E}\left[\|Z_{k\eta}^\eta - \eta \cdot \nabla U(Z_{k\eta}^\eta)\|^2\right] = \mathbb{E}\left[\|Z_{k\eta}^\eta\|^2\right] - 2\eta \cdot \mathbb{E}\left[\langle Z_{k\eta}^\eta, \nabla U(Z_{k\eta}^\eta)\rangle\right] + \eta^2 \cdot \mathbb{E}\left[\|\nabla U(Z_{k\eta}^\eta)\|^2\right]$$
$$\leq (1 - 2\alpha\eta + 2\eta^2 L^2) \cdot \mathbb{E}\left[\|Z_{k\eta}^\eta\|^2\right] + 2\eta\beta + 2\eta^2 L^2 \cdot \|x^*\|^2,$$

where $x^*$ is a stationary point of $U(\cdot)$. Hence, based on (C.37), we have

$$\mathbb{E}\left[\|Z_{(k+1)\eta}^\eta\|^2\right] \leq (1 - 2\alpha\eta + 2\eta^2 L^2) \cdot \mathbb{E}\left[\|Z_{k\eta}^\eta\|^2\right] + 2(\beta + d\tau_2)\eta + 2\eta^2 L^2\|x^*\|^2. \tag{C.38}$$

Note that when $\eta \in (0, \alpha/L^2)$, we have $1 - 2\alpha\eta + 2\eta^2 L^2 < 1$. Hence, according to (C.38), there exists a constant $\delta_1(d, \tau_2, L, \alpha, \beta)$ such that

$$\sup_{k \geq 0} \mathbb{E}\left[\|Z_{k\eta}^\eta\|^2\right] \leq \delta_1(d, \tau_2, L, \alpha, \beta),$$

which concludes the proof of Lemma C.2. $\square$

