# OpenReview forum: "ACCELERATING NONCONVEX LEARNING VIA REPLICA EXCHANGE LANGEVIN DIFFUSION"
_ICLR.cc/2019/Conference_

### Official Review · AnonReviewer1 · 2018-11-04
**well written.**

**Rating:** 6
**Confidence:** 4

**Review:**

PROS:
- The text is very well written, with a good balance between mathematical details and intuitions.
- I really like the high-level description of the algorithms and proof techniques

CONS:
to be completely honest, I am not sure I have learnt anything new from the paper.
1) the proof techniques are very standard
2) although there must be some small innovations, I thought that all the results had more or less been proven by Dupuis and co-authors:
a. large deviation principles
b. the larger the swapping rate, the better (which motivated Dupuis & al to consider the infinite swapping limit.)

and
c. Bakri & al methodology to prove convergence relying on the carre du champ is by now very standard and the proofs of the paper are only minor adaptations.

I must probably be missing something, and I encourage the authors to clarify what the main novelties are when compared to the several papers by Dupuis & al.

REMARKS:
1) I do not really understand the emphasis on optimisation while all the proofs are related to the convergence to the stationary distributions.

---

> ### Author Response · Authors · 2018-11-20
> **Our emphasis is an alternative approach and discretization analysis.**
>
> We really appreciate your comments. The main purpose of this paper is to introduce a new method to solve global optimization problem via replica exchange Langevin diffusion. We quantify the acceleration effect from the viewpoint of continuous time process. Although this work is inspired from Dupuis's work, their setting is MCMC and they only investigate by large deviation. We quantify the acceleration effect by both large deviation and chi^2 divergence. Besides, the large deviation rate function in our paper is different with that of Dupuis's since we use an alternative approach. We choose such a form of rate function because it is connected to the Dirichlet form, and hence, the convergence of chi^2 divergence. We acknowledge that our analysis tools is standard and not fancy in mathematics. However, this is not a mathematics conference after all. One of our contribution is applying standard mathematical tools to a specific machine learning problem. Finally, another contribution is that we propose a discretized algorithm. Although Dupuis& et.al's work establishes beautiful and complicated mathematical theory for replica exchange Langevin diffusions, they does not consider the discretization at all. In practice, we can only use the discretized one instead of the ideal continuous process to solve problems. Our theory quantifies the discretization error and convergence rate and hence, ensures the validity to use the discretized algorithm.

---

### Official Review · AnonReviewer3 · 2018-11-05
**Proof that replica exchange accelerates convergence in Langevin dynamics**

**Rating:** 7
**Confidence:** 4

**Review:**

This paper gives a theoretical analysis of an interesting statistical physics technique known as replica exchange. The basic idea is that Langevin dynamics at low temperature is slow to converge, and that one could potentially boost the convergence by alternating between low and high temperature. At the extreme one could imagine running in parallel a random search and a gradient descent, and ``teleporting" the gradient descent algorithm whenever the random search algorithm finds a point with better value. This makes a lot of sense and it is nice to see a theoretical analysis of this. The mathematics are sound, but I do not know whether it is an appropriate submission for ICLR.

One comment from the math side: it would be interesting (albeit probably difficult) to study kappa in (3.10) as a function of a. In particular at face value it looks like one only benefits from taking a larger, so why not study the limiting behavior of a->infty? What is the limiting value of kappa? Can you perform those calculations in the convex case at least?

---

> ### Author Response · Authors · 2018-11-22
> **Infinity swapping is not a trivial extension.**
>
> We really appreciate your comments. Replica exchange Langevin diffusion is widely used in classic MCMC over the years. Our work also uses this methodology in the setting of nonconvex optimization problem, which arises in many machine learning applications such as training neural networks. There are also many interesting questions in this direction, for example, how to choose the best temperature based on the structure of specific problems. That is why we still submit it for ICLR.
> As for the comments on math side. First, when a->infty, the exchange process should be defined in another way. Our current definition, which swapping particles with some rate, is only valid for finite a. This extension is not totally trivial and in Dupuis&et. al's work, some results are established. In our paper, we only discuss finite swap rate. This brings convenience for the discussion of discretization error. Otherwise, we need to use a different approach to analyze. Moreover, we point out that in discretization, the swapping intensity a should be smaller than the step size. This also reflects the nontrivial connection between infinity swapping and discretization.
> Second, the kappa in (3.10) is related to the Poincare inequality and it is also a lower estimate of the spectral gap of Markov process. Kappa is can be defined as the solution of a variational problem involved Dirichlet form. However, although our result shows that swapping boosts the Dirichlet form, we still cannot obtain an analytical formula of kappa depending on a, since the variational problem makes this relation extremely complicated. Even in the field of pure math, it is still very hard to obtain an explicit formula of kappa for a general Markov process. However, for this special case, we will keep trying to solve it in the future.

---

### Official Review · AnonReviewer2 · 2018-11-06
**well written but not weak results**

**Rating:** 4
**Confidence:** 4

**Review:**

The paper considers 'replica exchange' Langevin dynamics. These methods are very popular among practitioners, and developing some theory backing the empirical successes is an important goal.
Unfortunately this paper offers only weak results.
- The first 6 pages set up the general formalism. This is textbook material adapted to the current problem.
- Page 7 offers a result (expression for the Dirichlet form), which is hardly more than an exercise for anybody familiar with Markov Chains theory.
- Page 8 gives a Poincare inequality. Again, this follows from known results. More importantly: (1) It does not show any advantage of replica exchange over standard dynamics; (2) It does not provide any quantitative insight for high-dimensional problems.
- Similar comments hold for the following pages. They are an exercise in applying standard formalism to this problem, without really showing any significative advantage of replica exchange.

---

> ### Author Response · Authors · 2018-11-26
> **Standard analysis but nontrivial results.**
>
> We appreciate your valuable comments. As you have said, these methods are popular in practice and achieve good performance. However, most of them are done in the setting of MCMC, and people rarely use them in nonconvex optimization. One contribution of this paper is to apply these techniques to optimization problems.
>
> Our paper also tries to understand the acceleration effect of replica exchange. We quantify it in both LDP and the convergence of chi^2 divergence. Although in Dupuis’s work, he also quantifies the acceleration effect via LDP, the LDP theory we use in this paper is different from that. Specifically, his approach is based on the LDP variational theory, and ours is based on the theory of Donsker-Varadhan. As a result, our rate function has different form from his. In our paper, we also analyze the acceleration in the convergence of chi^2 divergence. It is a new perspective and not discussed by Dupuis. We emphasize that LDP and chi^2 divergence are two different approached to quantify convergence. They have different meanings. The first one characterizes the decay rate of the probability that the empirical measures deviate from the stationary measure and the second one characterizes the decay rate of the discrepancy between the transit distributions and limiting distribution.  Although the theory of LDP and convergence of chi^2 divergence for a general Markov process are well established and standard, to the best of our knowledge, our paper is the first to apply these tools in this specific problem.
>
> In our paper, one contribution is that we demonstrate the acceleration effect of replica exchange mathematically. We first show that the LDP rate function is boosted by replica exchange. Dupuis’s work includes similar results but in a different form. We also show that the derivative of chi^2 divergence is boosted. Specifically, we demonstrate that a strict positive term caused by the replica exchange is added, if the density ratio between current distribution and limiting distribution is not symmetric. We say that a function is symmetric if we swap the positions of variables, the function value does not change. In this case, the derivative of chi^2 divergence is strictly boosted, and hence, the convergence is accelerated strictly. It reflects the benefits of replica exchange. To the best of our knowledge, this phenomenon has never been observed by previous literature, including Dupuis’s paper. We think it is interesting and useful.
>
> Another contribution of our paper is the discretization algorithm. In practice, it is impossible to simulate the continuous process directly, and discretization is necessary. To the best of our knowledge, no one has discussed the discretization of replica exchange Langevin diffusion before. Our paper is the first one to analyze the discretization theoretically. In this paper, we establish the linear convergence rate for the discretization error, which is highly trivial since the process has state-dependent jumps. This result, combined with the acceleration effect, justifies the empirical success of the replica exchange Langevin diffusion in practice.

---

### Meta-Review · Area_Chair1 · 2018-12-16
**Interesting contribution, even if the analysis is mostly standard**

**Confidence:** 3
**Recommendation:** Accept (Poster)

**Metareview:**

The main criticisms were around novelty: that the analysis is rather standard. Given that all the reviewers agreed the paper is well written, I'm inclined to think the paper will be a useful contribution to the literature. The authors also highlight the analysis of the discretization, which seems to be missed by the most critical reviewer. I would suggest to the reviewers that they use the criticisms to rework the paper's introduction, to better explain which parts of the work are novel and which parts are standard. I would also suggest that standard background be moved to the appendix so that it is there for the nonexpert, while making the body of the work more focused on the novel aspects.